# Striatal action-value neurons reconsidered

Lotem Elber-Dorozko[1]*, Yonatan Loewenstein[1,2,3]

[1]The Edmond & Lily Safra Center for Brain Sciences, The Hebrew University of Jerusalem, Jerusalem, Israel; [2]Department of Neurobiology, The Alexander Silberman Institute of Life Sciences, The Hebrew University of Jerusalem, Jerusalem, Israel; [3]The Federmann Center for the Study of Rationality, The Hebrew University of Jerusalem, Jerusalem, Israel

**Abstract** It is generally believed that during economic decisions, striatal neurons represent the values associated with different actions. This hypothesis is based on studies, in which the activity of striatal neurons was measured while the subject was learning to prefer the more rewarding action. Here we show that these publications are subject to at least one of two critical confounds. First, we show that even weak temporal correlations in the neuronal data may result in an erroneous identification of action-value representations. Second, we show that experiments and analyses designed to dissociate action-value representation from the representation of other decision variables cannot do so. We suggest solutions to identifying action-value representation that are not subject to these confounds. Applying one solution to previously identified action-value neurons in the basal ganglia we fail to detect action-value representations. We conclude that the claim that striatal neurons encode action-values must await new experiments and analyses.

DOI: https://doi.org/10.7554/eLife.34248.001

*For correspondence:
lotem.elber@mail.huji.ac.il

Competing interests: The authors declare that no competing interests exist.

There is a long history of operant learning experiments, in which a subject, human or animal, repeatedly chooses between actions and is rewarded according to its choices. A popular theory posits that the subject's decisions in these tasks utilize estimates of the different *action-values*. These action-values correspond to the expected reward associated with each of the actions, and actions associated with a higher estimated action-value are more likely to be chosen (*Sutton and Barto, 1998*). In recent years, there is a lot of interest in the neural mechanisms underlying this computation (*Louie and Glimcher, 2012*; *Schultz, 2015*). In particular, based on electrophysiological, functional magnetic resonance imaging (fMRI) and intervention experiments, it is now widely accepted that a population of neurons in the striatum represents these action-values, adding sway to this action-value theory (*Cai et al., 2011*; *FitzGerald et al., 2012*; *Funamizu et al., 2015*; *Guitart-Masip et al., 2012*; *Her et al., 2016*; *Ito and Doya, 2009*; *2015a*; *Ito and Doya, 2015b*; *Kim et al., 2013*; *Kim et al., 2009*; *Kim et al., 2012*; *2007*; *Lau and Glimcher, 2008*; *Lee et al., 2015*; *Samejima et al., 2005*; *Stalnaker et al., 2010*; *Tai et al., 2012*; *Wang et al., 2013*; *Wunderlich et al., 2009*). Here we challenge the evidence for action-value representation in the striatum by describing two major confounds in the interpretation of the data that have not yet been successfully addressed.

To identify neurons that represent the values the subject associates with the different actions, researchers have searched for neurons whose firing rate is significantly correlated with the average reward associated with exactly one of the actions. There are several ways of defining the average reward associated with an action. For example, the average reward can be defined by the reward schedule, for example, the probability of a reward associated with the action. Alternatively, one can adopt the subject's perspective, and use the subject-specific history of rewards and actions in order to estimate the average reward. In particular, the Rescorla–Wagner model (equivalent to the standard ones-state Q-learning model) has been used to estimate action-values (*Kim et al., 2009*;

*Samejima et al., 2005*). In this model, the value associated with an action $i$ in trial $t$, termed $Q_i(t)$, is an exponentially-weighted average of the rewards associated with this action in past trials:

$$Q_i(t+1) = Q_i(t) + \alpha(R(t) - Q_i(t)) \ \text{ if } a(t) = i \tag{1}$$

$$Q_i(t+1) = Q_i(t) \ \text{ if } a(t) \neq i$$

where $a(t)$ and $R(t)$ denote the choice and reward in trial $t$, respectively, and $\alpha$ is the learning rate.

The model also posits that in a two-alternative task, the probability of choosing an action is a sigmoidal function, typically softmax, of the difference of the action-values (see also [*Shteingart and Loewenstein, 2014*]):

$$\Pr(a(t) = 1) = \frac{1}{1 + e^{-\beta(Q_1(t) - Q_2(t))}} \tag{2}$$

where $\beta$ is a parameter that determines the bias towards the action associated with the higher action-value. The parameters of the model, $\alpha$ and $\beta$, can be estimated from the behavior, allowing the researchers to compute $Q_1$ and $Q_2$ on a trial-by-trial basis.

In principle, one can identify the neurons that represent an action-value by identifying neurons for which the regression of the trial-by-trial spike count on one of the variables $Q_i(t)$ is statistically significant. Using this framework, electrophysiological studies have found that the firing rate of a substantial fraction of striatal neurons (12–40% for different significance thresholds) is significantly correlated with an action-value. These and similar results were considered as evidence that neurons in the striatum represent action-values (*Funamizu et al., 2015*; *Her et al., 2016*; *Ito and Doya, 2015a*; *Ito and Doya, 2015b*; *Kim et al., 2013*; *Kim et al., 2009*; *Lau and Glimcher, 2008*; *Samejima et al., 2005*).

In this paper we conduct a systematic literature search and conclude that the literature has, by and large, ignored two major confounds in this and in similar analyses. First, it is well-known that spurious correlations can emerge in correlation analysis if both variables have temporal correlations (*Granger and Newbold, 1974*; *Phillips, 1986*). Here we show that neurons can be erroneously classified as representing action-values when their firing rates are weakly temporally correlated. Second, it is also well-known that lack of a statistically significant result in the analysis does not imply lack of correlation. Because in standard analyses neurons are classified as representing action-values if they have a significant regression coefficient on *exactly* one action-value and because decision variables such as policy are correlated with both action-values, neurons representing other decision variables may be misclassified as representing action-values. We propose different approaches to address these issues. Applying one of them to recordings from the basal ganglia, we fail to identify any action-value representation there. Thus, we conclude that the hypothesis that striatal neurons represent action-values still remains to be tested by experimental designs and analyses that are not subject to these confounds. In the Discussion we address additional conceptual issues with identifying such a representation.

This paper discusses methodological problems that may also be of relevance in other fields of biology in general and neuroscience in particular. Nevertheless, the focus of this paper is a single scientific claim, namely, that action-value representation in the striatum is an established fact. Our criticism is restricted to the representation of action-values, and we do not make any claims regarding the possible representations of other decision variables, such as policy, chosen-value or reward-prediction-error. This we leave for future studies. Moreover, we do not make any claims about the possible representations of action-values elsewhere in the brain, although our results suggest caution when looking for such representations.

The paper is organized in the following way. We commence by describing a standard method for identifying action-value neurons. Next, we show that this method erroneously classifies simulated neurons, whose activity is temporally correlated, as representing action-values. We show that this confound brings into question the conclusion of many existing publications. Then, we propose different methods for identifying action-value neurons, that overcome this confound. Applying such a method to basal ganglia recordings, in which action-value neurons were previously identified, we fail to conclusively detect any action-value representations. We continue by discussing the second confound: neurons that encode the policy (the probability of choice) may be erroneously classified as

representing action-value, even when the policy is the result of learning algorithms that are devoid of action-value calculation. Then we discuss a possible solution to this confound.

## Results

### Identifying action-value neurons

We commence by examining the standard methods for identifying action-value neurons using a simulation of an operant learning experiment. We simulated a task, in which the subject repeatedly chooses between two alternative actions, which yield a binary reward with a probability that depends on the action. Specifically, each session in the simulation was composed of four blocks such that the probabilities of rewards were fixed within a block and varied between the blocks. The probabilities of reward in the blocks were (0.1,0.5), (0.9,0.5), (0.5,0.9) and (0.5,0.1) for actions 1 and 2, respectively (*Figure 1A*). The order of blocks was random and a block terminated when the more rewarding action was chosen more than 14 times within 20 consecutive trials (*Ito and Doya, 2015a*; *Samejima et al., 2005*).

To simulate learning behavior, we used the Q-learning framework (*Equations 1 and 2* with $\alpha = 0.1$ and $\beta = 2.5$ (taken from distributions reported in [*Kim et al., 2009*]) and initial conditions $Q_i(1) = 0.5$). As demonstrated in *Figure 1A*, the model learned: the probability of choosing the more rewarding alternative increased over trials (black line). To model the action-value neurons, we simulated neurons whose firing rate is a linear function of one of the two Q-values and whose spike count in a 1 sec trial is randomly drawn from a corresponding Poisson distribution (see Materials and methods). The firing rates and spike counts of two such neurons, representing action-values 1 and 2, are depicted in *Figure 1B* in red and blue, respectively.

One standard method for identifying action-value neurons is to compare neurons' spike counts after learning, at the end of the blocks (horizontal bars in *Figure 1B*). Considering the red-labeled Poisson neuron, the spike count in the last 20 trials of the second block, in which the probability of reward associated with action 1 was 0.9, was significantly higher than that count in the first block, in which the probability of reward associated with action 1 was 0.1 (p<0.01; rank sum test). By contrast, there was no significant difference in the spike counts between the third and fourth blocks, in which the probability of reward associated with action 1 was equal (p=0.91; rank sum test). This is consistent with the fact that the red-labeled neuron was an action 1-value neuron: its firing rate was a linear function of the value of action 1 (*Figure 1B*, red) Similarly for the blue labeled neuron, the spike counts in the last 20 trials of the first two blocks were not significantly different (p=0.92; rank sum test), but there was a significant difference in the counts between the third and fourth blocks (p<0.001; rank sum test). These results are consistent with the probabilities of reward associated with action 2 and the fact that in our simulations, this neuron's firing rate was modulated by the value of action 2 (*Figure 1B*, blue).

This approach for identifying action-value neurons is limited, however, for several reasons. First, it considers only a fraction of the data, the last 20 trials in a block. Second, action-value neurons are not expected to represent the block average probabilities of reward. Rather, they will represent a subjective estimate, which is based on the subject-specific history of actions and rewards. Therefore, it is more common to identify action-value neurons by regressing the spike count on subjective action-values, estimated from the subject's history of choices and rewards (*Funamizu et al., 2015*; *Ito and Doya, 2015a*; *Ito and Doya, 2015b*; *Kim et al., 2009*; *Lau and Glimcher, 2008*; *Samejima et al., 2005*). Note that when studying behavior in experiments, we have no direct access to these estimated action-values, in particular because the values of the parameters $\alpha$ and $\beta$ are unknown. Therefore, following common practice, we estimated the values of $\alpha$ and $\beta$ from the model's sequence of choices and rewards using maximum likelihood, and used the estimated learning rate ($\alpha$) and the choices and rewards to estimate the action-values (thin lines in *Figure 1C*, see Materials and methods). These estimates were similar to the true action-value, which underlay the model's choice behavior (thick lines in *Figure 1C*).

Next, we regressed the spike count of each simulated neuron on the two estimated action-values from its corresponding session. As expected, the t-value of the regression coefficient of the red-labeled action 1-value neuron was significant for the estimated $Q_1$ ($t_{182}(Q_1) = 4.05$) but not for the estimated $Q_2$ ($t_{182}(Q_2) = -0.27$). Similarly, the t-value of the regression coefficient of the blue-labeled

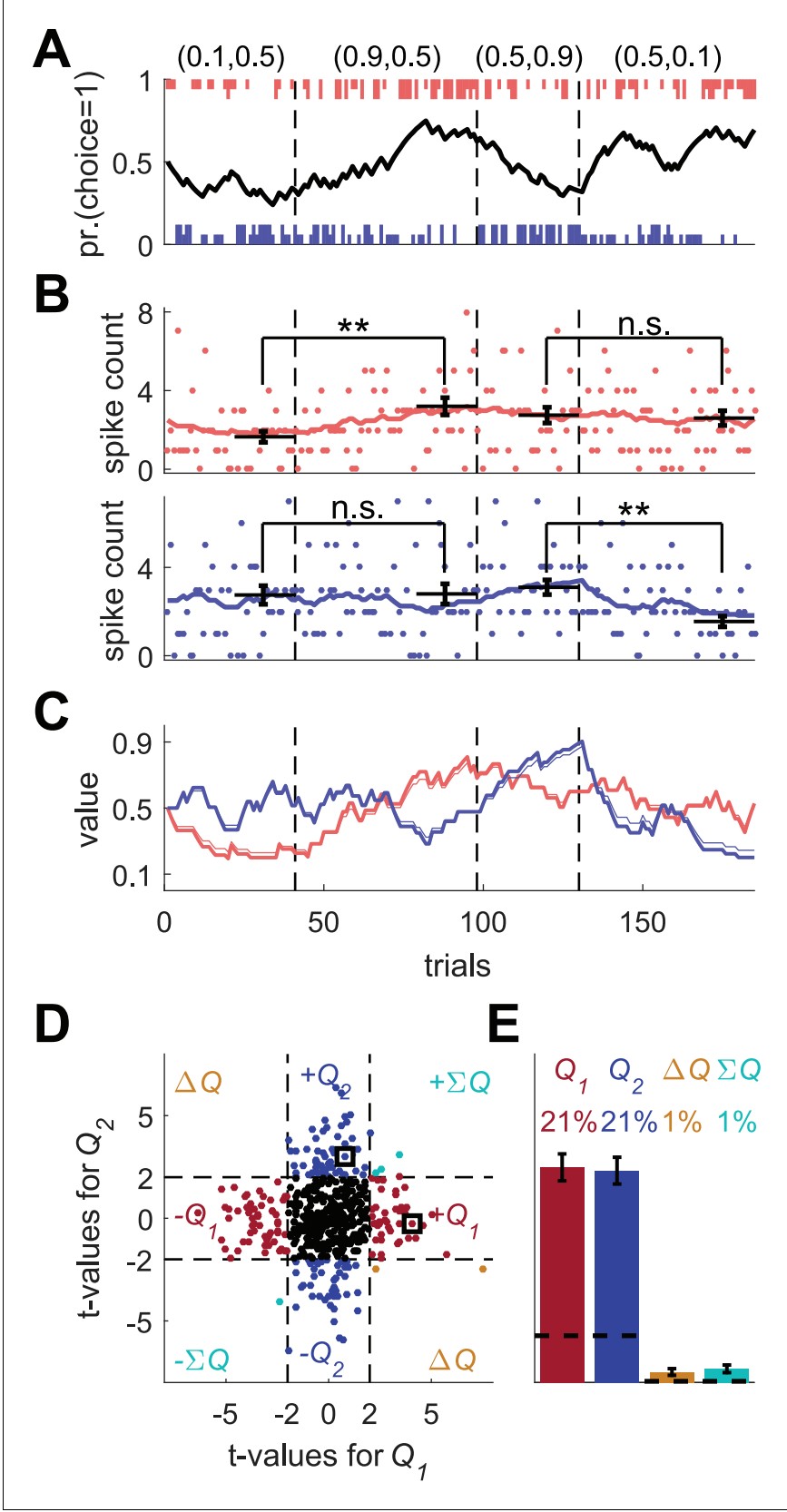

**Figure 1.** Model of action-value neurons. (**A**) Behavior of the model in an example session, composed of four blocks (separated by dashed vertical lines). The probabilities of reward for choosing actions 1 and 2 are denoted

*Figure 1 continued on next page*

*Figure 1 continued*

by the pair of numbers above the block. Black line denotes the probability of choosing action 1; vertical lines denote choices in individual trials, where red and blue denote actions 1 and 2, respectively, and long and short lines denote rewarded and unrewarded trials, respectively. (B) Neural activity. Firing rate (line) and spike-count (dots) of two example simulated action-value neurons in the session depicted in (A). The red and blue-labeled neurons represent $Q_1$ and $Q_2$, respectively. Black horizontal lines denote the mean spike count in the last 20 trials of the block. Error bars denote the standard error of the mean. The two asterisks denote p<0.01 (rank sum test). (C) Values. Thick red and blue lines denote $Q_1$ and $Q_2$, respectively. Note that the firing rates of the two neurons in (B) are a linear function of these values. Thin red and blue lines denote the estimates of $Q_1$ and $Q_2$, respectively, based on the choices and rewards in (A). The similarity between the thick and thin lines indicates that the parameters of the model can be accurately estimated from the behavior (see also Materials and methods). (D) and (E) Population analysis. (D) Example of 500 simulated action-value neurons from randomly chosen sessions. Each dot corresponds to a single neuron and the coordinates correspond to the t-values of the regression of the spike counts on the estimated values of the two actions. Dashed lines at t=2 denote the significance boundaries. Color of dots denote significance: dark red and blue denote a significant regression coefficient on exactly one estimated action-value, action 1 or action 2, respectively; light blue – significant regression coefficients on both estimated action-values with similar signs ($\Sigma Q$)); orange - significant regression coefficients on both estimated action-values with opposite signs ($\Delta Q$)); Black – no significant regression coefficients. The two simulated neurons in (B) are denoted by squares. (E) Fraction of neurons in each category, estimated from 20,000 simulated neurons in 1,000 sessions. Error bars denote the standard error of the mean. Dashed lines denote the naïve expected false positive rate from the significance threshold (see Materials and methods).

DOI: https://doi.org/10.7554/eLife.34248.002

action 2-value neuron was significant for the estimated $Q_2$ $(t_{182}(Q_2) = 3.05)$ but not for the estimated $Q_1$ $(t_{182}(Q_1) = 0.78)$.

A population analysis of the t-values of the two regression coefficients is depicted in *Figure 1D, E*. As expected, a substantial fraction (42%) of the simulated neurons were identified as action-value neurons. Only 2% of the simulated neurons had significant regression coefficients with both action-values. Such neurons are typically classified as state ($\Sigma Q$) or policy (also known as preference) ($\Delta Q$) neurons, if the two regression coefficients have the same or different signs, respectively (*Ito and Doya, 2015a*). Note that despite the fact that by construction, all neurons were action-value neurons, not all of them were detected as such by this method. This failure occurred for two reasons. First, the estimated action-values are not identical to the true action-values, which determine the firing rates. This is because of the finite number of trials and the stochasticity of choice (note the difference, albeit small, between the thin and thick lines in *Figure 1C*). Second and more importantly, the spike count in a trial is only a noisy estimate of the firing rate because of the Poisson generation of spikes.

Several prominent studies have implemented the methods we described in this section and reported that a substantial fraction (10–40% depending on significance threshold) of striatal neurons represent action-values (*Ito and Doya, 2015a*; *Ito and Doya, 2015b*; *Samejima et al., 2005*). In the next two sections we show that these methods, and similar methods employed by other studies (*Cai et al., 2011*; *FitzGerald et al., 2012*; *Funamizu et al., 2015*; *Guitart-Masip et al., 2012*; *Her et al., 2016*; *Ito and Doya, 2009*; *Kim et al., 2013*; *Kim et al., 2009*; *Kim et al., 2012*; *2007*; *Lau and Glimcher, 2008*; *Stalnaker et al., 2010*; *Wang et al., 2013*; *Wunderlich et al., 2009*) are all subject to at least one of two major confounds.

## Confound 1 – temporal correlations
### Simulated random-walk neurons are erroneously classified as action-value neurons

The red and blue-labeled neurons in *Figure 1D* were classified as action-value neurons because their t-values were improbable under the null hypothesis that the firing rate of the neuron is not modulated by action-values. The significance threshold ($t = 2$) was computed assuming that trials are independent in time. To see why this assumption is essential, we consider a case in which it is violated. *Figure 2A* depicts the firing rates and spike counts of two simulated Poisson neurons, whose firing rates follow a bounded Gaussian random-walk process:

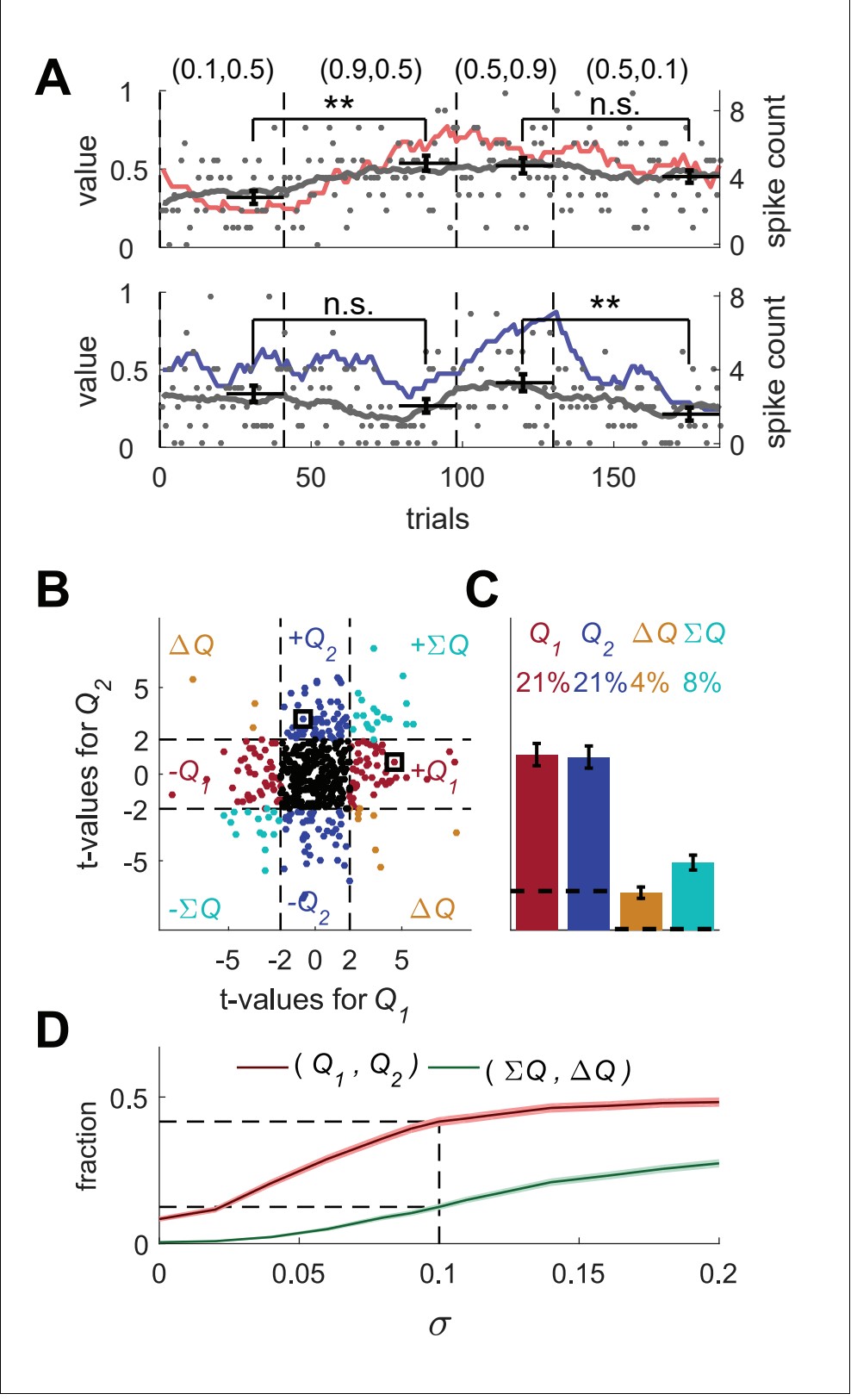

**Figure 2.** Erroneous detection of action-value representation in random-walk neurons. (**A**) Two example random-walk neurons that appear as if they represent action-values. The red (top) and blue (bottom) lines denote the estimated action-values 1 and 2, respectively that were depicted in *Figure 1C*. Gray lines and gray dots denote

*Figure 2 continued on next page*

*Figure 2 continued*

the firing rates and the spike counts of two example random-walk neurons that were randomly assigned to this simulated session. Black horizontal lines denote the mean spike count in the last 20 trials of each block. Error bars denote the standard error of the mean. The two asterisks denote p<0.01 (rank sum test). (B) and (C) Population analysis. Each random-walk neuron was regressed on the two estimated action-values, as in *Figure 1D and E*. Numbers and legend are the same as in *Figure 1D and E*. The two random-walk neurons in (A) are denoted by squares in (B). Dashed lines in (B) at t=2 denote the significance boundaries. Dashed lines in (C) denote the naïve expected false positive rate from the significance threshold (see Materials and methods). (D) Fraction of random-walk neurons classified as action-value neurons (red), and classified as state neurons ($\Sigma Q$) or policy neurons ($\Delta Q$) (green) as a function of the magnitude of the diffusion parameter of random-walk ($\sigma$). Light red and light green are standard error of the mean. Dashed lines denote the results for $\sigma$=0.1, which is the value of the diffusion parameter used in (A)-(C). Initial firing rate for all neurons in the simulations is $f(1) = 2.5$Hz.

DOI: https://doi.org/10.7554/eLife.34248.003

The following figure supplements are available for figure 2:

**Figure supplement 1.** Erroneous detection of action-value representation in a model with covariance based synaptic plasticity.
DOI: https://doi.org/10.7554/eLife.34248.004
**Figure supplement 2.** Erroneous detection of action-value neurons in unrelated experiments.
DOI: https://doi.org/10.7554/eLife.34248.005
**Figure supplement 3.** Erroneous detection of unrelated action-value representations in basal ganglia neurons.
DOI: https://doi.org/10.7554/eLife.34248.006
**Figure supplement 4.** Spike count permutation (as in [*Kim et al., 2009*]) does not resolve the temporal correlations confound.
DOI: https://doi.org/10.7554/eLife.34248.007
**Figure supplement 5.** Autoregressive coefficients do not resolve the temporal correlations confound.
DOI: https://doi.org/10.7554/eLife.34248.008
**Figure supplement 6.** Regression on reward probabilities does not resolve the temporal correlations confound.
DOI: https://doi.org/10.7554/eLife.34248.009
**Figure supplement 7.** Detrending analysis does not resolve the temporal correlations confound.
DOI: https://doi.org/10.7554/eLife.34248.010
**Figure supplement 8.** Unbiased classification of action-value neurons does not resolve the temporal correlations confound.
DOI: https://doi.org/10.7554/eLife.34248.011
**Figure supplement 9.** Random intermingling of estimated action-values does not resolve the temporal correlations confound.
DOI: https://doi.org/10.7554/eLife.34248.012
**Figure supplement 10.** Increasing the number of blocks does not resolve the temporal correlations confound.
DOI: https://doi.org/10.7554/eLife.34248.013

$$f(t+1) = [f(t) + z(t)]_+ \tag{3}$$

where $f(t)$ is the firing rate in trial $t$ (we consider epochs of 1 second as 'trials'), $z(t)$ is a diffusion variable, randomly and independently drawn from a normal distribution with mean 0 and variance $\sigma^2 = 0.01$ and $[x]_+$ denotes a linear-threshold function, $[x]_+ = x$ if $x \geq 0$ and 0 otherwise.

These random-walk neurons are clearly not action-value neurons. Nevertheless, we tested them using the analyses depicted in *Figure 1*. To that goal, we randomly matched the trials in the simulation of the random-walk neurons (completely unrelated to the task) to the trials in the simulation depicted in *Figure 1A*. Then, we considered the spike counts of the random-walk neurons in the last 20 trials of each of the four blocks in *Figure 1A* (block being defined by the simulation of learning and is unrelated to the activity of the random-walk neurons). Surprisingly, when considering the top neuron in *Figure 2A* and utilizing the same analysis as in *Figure 1B*, we found that its spike count differed significantly between the first two blocks (p<0.01, rank sum test) but not between the last two blocks (p=0.28, rank sum test), similarly to the simulated action 1-value neuron of *Figure 1B* (red). Similarly, the spike count of the bottom random-walk neuron matched that of a simulated action 2-value neuron (compare with the blue-labeled neuron in *Figure 1B*; *Figure 2A*).

Moreover, we regressed each vector of spike counts for 20,000 random-walk neurons on randomly matched estimated action-values from *Figure 1E* and computed the t-values (*Figure 2B*). This analysis erroneously classified 42% of these random-walk neurons as action-value neurons (see *Figure 2C*). In particular, the top and bottom random-walk neurons of *Figure 2A* were identified as action-value neurons for actions 1 and 2, respectively (squares in *Figure 2B*).

To further quantify this result, we computed the fraction of random-walk neurons erroneously classified as action-value neurons as a function of the diffusion parameter $\sigma$ (*Figure 2D*). When $\sigma=0$, the spike counts of the neurons in the different trials are independent and the number of random-walk neurons classified as action-value neurons is slightly less than 10%, the fraction expected by chance from a significance criterion of 5% and two statistical tests, corresponding to the two action-values. The larger the value of $\sigma$, the higher the probability that a random-walk neuron will pass the selection criterion for at least one action-value and thus be erroneously classified as an action-value, state or policy neuron.

The excess action-value neurons in *Figure 2* emerged because the significance boundary in the statistical analysis was based on the assumption that the different trials are independent from each other. In the case of a regression of a random-walk process on an action-value related variable, this assumption is violated. The reason is that in this case, both predictor (action-value) and the dependent variable (spike count) slowly change over trials, the former because of the learning and the latter because of the random drift. As a result, the statistic, which relates these two signals, is correlated between trials, violating the independence-of-trials assumption of the test. Because of these dependencies, the expected variance of the statistic (be it average spike count in 20 trials or the regression coefficient), which is calculated under the independence-of-trials assumption, is an underestimate of the actual variance. Therefore, the fraction of random-walk neurons classified as action-value neurons increases with the magnitude of the diffusion, which is directly related to the magnitude of correlations between spike counts in proximate trials (*Figure 2D*). The phenomenon of spurious significant correlations in time-series with temporal correlations has been described previously in the field of econometrics and a formal discussion of this issue can be found in (*Granger and Newbold, 1974*; *Phillips, 1986*).

## Is this confound relevant to the question of action-value representation in the striatum?

### Is a random-walk process a good description of striatal neurons' activity?

The Gaussian random-walk process is just an example of a temporally correlated firing rate and we do not argue that the firing rates of striatal neurons follow such a process. However, any other type of temporal correlations, for example, oscillations or trends, will violate the independence-of-trials assumption, and may lead to the erroneous classification of neurons as representing action-values. Such temporal correlations can also emerge from stochastic learning. For example, in *Figure 2—figure supplement 1* we consider a model of operant leaning that is based on covariance based synaptic plasticity (*Loewenstein, 2008*; *Loewenstein, 2010*; *Loewenstein and Seung, 2006*; *Neiman and Loewenstein, 2013*) and competition (*Bogacz et al., 2006*). Because such plasticity results in slow changes in the firing rates of the neurons, applying the analysis of *Figure 1E* to our simulations results in the erroneous classification of 43% of the simulated neurons as representing action-values. This is despite the fact that action-values are not computed as part of this learning, neither explicitly or implicitly.

### Are temporal correlations in neural recordings sufficiently strong to affect the analysis?

To test the relevance of this confound to experimentally-recorded neural activity, we repeated the analysis of *Figure 2B,C* on neurons recorded in two unrelated experiments: 89 neurons from extracellular recordings in the motor cortex of an awake monkey (*Figure 2—figure supplement 2A–B*) and 39 auditory cortex neurons recorded intracellularly in anaesthetized rats (*Figure 2—figure supplement 2C–D*; [*Hershenhoren et al., 2014*]). We regressed the spike counts on randomly matched estimated action-values from *Figure 1E*. In both cases we erroneously classified neurons as representing action-value in a fraction comparable to that reported in the striatum (36 and 23%, respectively).

## Strong temporal correlations in the striatum

To test the relevance of this confound to striatal neurons, we considered previous recordings from neurons in the nucleus accumbens (NAc) and ventral pallidum (VP) of rats in an operant learning experiment (*Ito and Doya, 2009*) and regressed their spike counts on simulated, unrelated action-values (using more blocks and trials than in *Figure 1E*, see Figure legend). Note that although the recordings were obtained during an operant learning task, the action-values that we used in the regression were obtained from simulated experiments and were completely unrelated to the true experimental settings. Again, we erroneously classified a substantial fraction of neurons (43%) as representing action-values, a fraction comparable to that reported in the striatum (*Figure 2—figure supplement 3*).

## Haven't previous publications acknowledged this confound and successfully addressed it?

We conducted an extensive literature search to see whether previous studies have identified this confound and addressed it (see Materials and methods). Two studies noted that processes such as slow drift in firing rate may violate the independence-of-trials assumption of the statistical tests and suggested unique methods to address this problem (*Kim et al., 2013*; *Kim et al., 2009*): one method (*Kim et al., 2009*) relied on permutation of the spike counts within a block (*Figure 2—figure supplement 4*, see Materials and methods) and another (*Kim et al., 2013*), used spikes in previous trials as predictors (*Figure 2—figure supplement 5*). However, both approaches still erroneously classify unrelated recorded and random-walk neurons as action-value neurons (*Figure 2—figure supplements 4* and *5*). The failure of both these approaches stems from the fact that a complete model of the learning-independent temporal correlations is lacking. As a result, these methods are unable to remove *all* the temporal correlations from the vector of spike-counts.

Our literature search yielded four additional methods that have been used to identify action-value neurons. However, as depicted in *Figure 2—figure supplement 6* (corresponding to the analyses in [*Ito and Doya, 2009*; *Samejima et al., 2005*]), *Figure 2—figure supplement 7* (corresponding to the analysis in [*Ito and Doya, 2015a*]), *Figure 2—figure supplement 8* (corresponding to the analysis in [*Wang et al., 2013*]) and *Figure 2—figure supplement 9* (corresponding to a trial design experiment in [*FitzGerald et al., 2012*]), all these additional methods erroneously classify neurons from unrelated recordings and random-walk neurons as action-value neurons in numbers comparable to those reported in the striatum (*Figure 2—figure supplement 6–9*). The fMRI analysis in (*FitzGerald et al., 2012*) focused on the difference between action-values rather than on the action-values themselves (see confound 2), and therefore we did not attempt to replicate it (and cannot attest to whether it is subject to the temporal correlations confound). We did, however, conduct the standard analysis on their unique experimental design - a trial-design experiment in which trials with different reward probabilities are randomly intermingled. Surprisingly, we erroneously detect action-value representation even when using this trial design (*Figure 2—figure supplement 9*). This erroneous detection occurs because in this analysis, the regression's predictors are estimated action-values, which are temporally correlated. From this example it follows that even trial-design experiments may still be subject to the temporal correlations confound.

## Some previous publications used more blocks. Shouldn't adding blocks solve the problem?

In *Figures 1* and *2* we considered a learning task composed of four blocks with a mean length of 174 trials (standard deviation 43 trials). It is tempting to believe that experiments with more blocks and trials (e.g., [*Ito and Doya, 2009*; *Wang et al., 2013*]) will be immune to this confound. The intuition is that the larger the number of trials, the less likely it is that a neuron that is not modulated by action-value (e.g., a random-walk neuron) will have a large regression coefficient on one of the action-values. Surprisingly, however, this intuition is wrong. In *Figure 2—figure supplement 10* we show that doubling the number of blocks, so that the original blocks are repeated twice, each time in a random order, does not decrease the fraction of neurons erroneously classified as representing action-values. For the case of random-walk neurons, it can be shown that, contrary to this intuition, the fraction of erroneously identified action-value neurons is expected to increase with the number of trials (*Phillips, 1986*). This is because the expected variance of the regression coefficients under the null hypothesis is inversely proportional to the degrees of freedom, which increase with the

number of trials. As a result, the threshold for classifying a regression coefficient as significant decreases with the number of trials.

## Possible solutions to the temporal correlations confound

The temporal correlations confound has been acknowledged in the fMRI literature, and several methods have been suggested to address it, such as 'prewhitening' (*Woolrich et al., 2001*). However, these methods require prior knowledge, or an estimate of the predictor-independent temporal correlations. Both are impractical for the slow time-scale of learning and therefore are not applicable in the experiments we discussed.

Another suggestion is to assess the level of autocorrelations between trials in the data and to use it to predict the expected fraction of erroneous classification of action-value neurons. However, using such a measure is problematic in the context of action-value representation because the autocorrelations relevant for the temporal correlations confound are those associated with the time-scale relevant for learning - tens of trials. Computing such autocorrelations in experiments of a few hundreds of trials introduces substantial biases (*Kohn, 2006*; *Newbold and Agiakloglou, 1993*). Moreover, even when these autocorrelations are computed, it is not clear exactly how they can be used to estimate the expected false positive rate for action-value classification.

Finally, it has been suggested that the temporal correlation confound can be addressed by using repeating blocks and removing neurons whose activity is significantly different in identical blocks (*Asaad et al., 2000*; *Mansouri et al., 2006*). We applied this method by applying a design in which the four blocks of *Figure 1* are repeated twice. However, even when this method was applied, a significant number of neurons were erroneously classified as representing action-values (Materials and methods).

We therefore propose two alternative approaches.

## Permutation analysis

Trivially, an action-value neuron (or any task-related neuron) should be more strongly correlated with the action-value of the experimental session, in which the neuron was recorded, than with action-values of other sessions (recorded in different days). We propose to use this requirement in a permutation test, as depicted in *Figure 3*. We first consider the two simulated action-value neurons of *Figure 1B*. For each of the two neurons, we computed the t-values of the regression coefficients of the spike counts on each of the estimated action-values in all possible sessions (see Materials and methods). *Figure 3A* depicts the two resulting distributions of t-values. As a result of the temporal correlations, the 5% significance boundaries (vertical dashed lines), which are defined to be exceeded by exactly 5% of t-values in each distribution, are substantially larger (in absolute value) than 2, the standard significance boundaries. On this analysis, a neuron is significantly correlated with an action-value if the t-value of the regression on the action-value from its corresponding session exceeds the significance boundaries derived from the regression of its spike count on all possible action-values.

Indeed, when considering the Top (red) simulated action 1-value neuron, we find that its spike count has a significant regression coefficient on the estimated $Q_1$ from its session (red arrow) but not on the estimated $Q_2$ (blue arrow). Importantly, because the significance boundary exceeds 2, this approach is less sensitive than the original one (*Figure 1*) and indeed, the regression coefficients of the Bottom simulated neuron (blue) do not exceed the significance level (red and blue arrows) and thus this analysis fails to identify it as an action-value neuron. Considering the population of simulated action-value neurons of *Figure 1*, this analysis identified 29% of the action-value neurons of *Figure 1* as such (*Figure 3B*, green), demonstrating that this analysis can identify action-value neurons. When considering the random-walk neurons (*Figure 2*), this method classifies only approximately 10% of the random-walk neurons as action-value neurons, as predicted by chance (*Figure 3B*, yellow). Similar results were obtained for the motor cortex and auditory cortex neurons (not shown).

## Permutation analysis of basal ganglia neurons

Importantly, this permutation method can also be used to reanalyze the activity of previously recorded neurons. To that goal, we considered the recordings reported in (*Ito and Doya, 2009*). The results of their model-free method (*Figure 2—figure supplement 6*) imply that

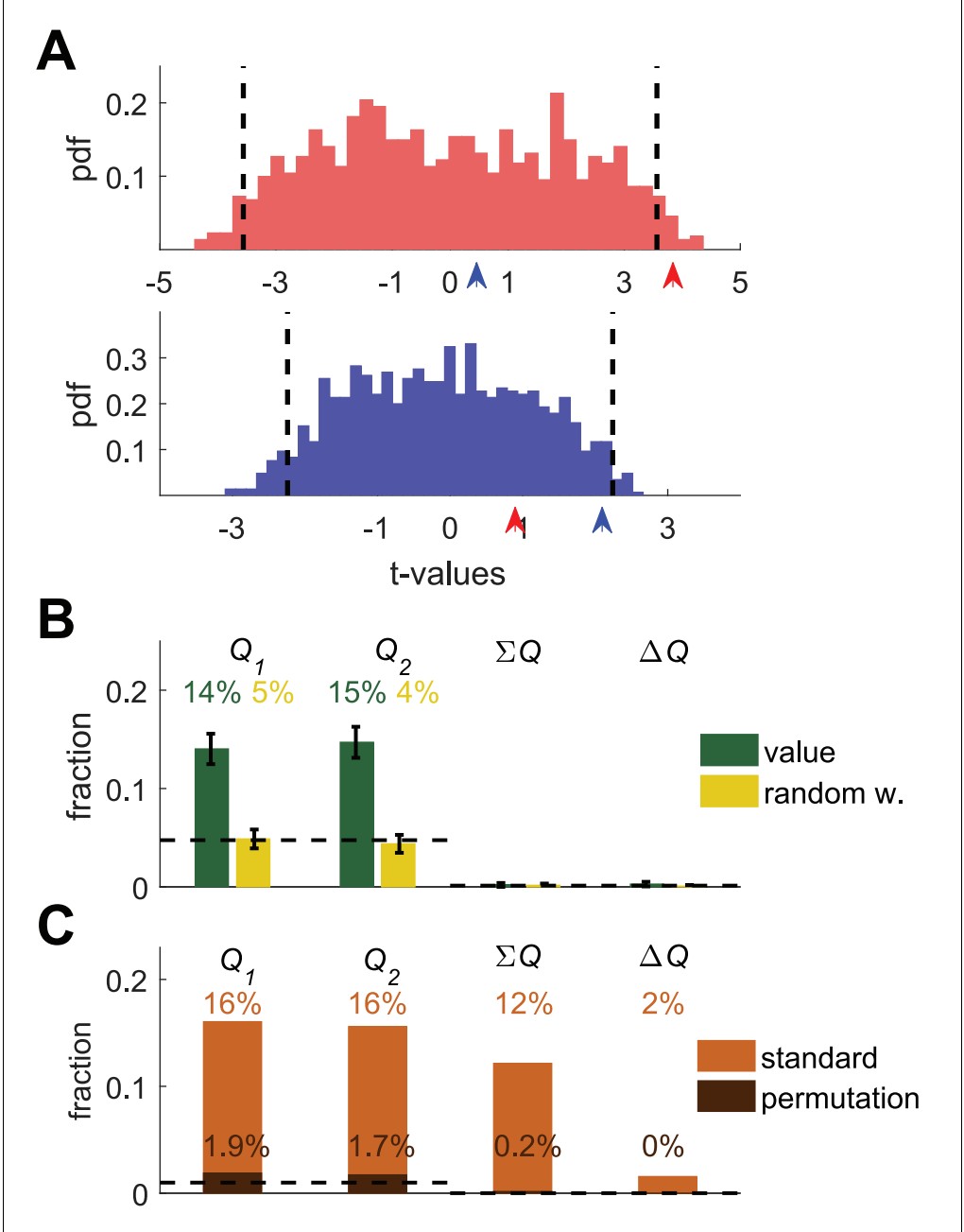

**Figure 3.** Permutation analysis. (**A**) Red and blue correspond to red and blue - labeled neurons in *Figure 1B*. Arrow-heads denote the t-values from the regressions on the estimated action-values from the session in which the neuron was simulated (depicted in *Figure 1A*). The red and blue histograms denote the t-values of the regressions of the spike-count on estimated action-values from *different* sessions in *Figure 1E* (Materials and methods). Dashed black lines denote the 5% significance boundary. In this analysis, the regression coefficient of neural activity on an action-value is significant if it exceeds these significance boundaries. Note that because of the temporal correlations, these significance boundaries are larger than ±2 (the significance boundaries in *Figure 1,2*). According to this permutation test the red-labeled but not the blue-labeled neuron is classified as an action-value neuron (**B**) Fraction of neurons classified in each category using the permutation analysis for the action-value neurons (green, *Figure 1*) and random-walk neurons (yellow, *Figure 2*).Dashed lines denote the naïve expected false positive rate from the significance threshold (Materials and methods). Error bars denote the standard error of the mean. The test correctly identifies 29% of actual action-value neurons as such, while classification of random-walk neurons was at chance level. Analysis was done on 10,080 action-value neurons and 10,077 random-walk neurons from 504 simulated sessions (**C**) Light orange, fraction of basal ganglia neurons from

*Figure 3 continued*

(*Ito and Doya, 2009*) classified in each category when regressing the spike count of 214 basal ganglia neurons in three different experimental phases on the estimated action-values associated with their experimental session. This analysis classified 32% of neurons as representing action-values. Dark orange, fraction of basal ganglia neurons classified in each category when applying the permutation analysis. This test classified 3.6% of neurons as representing action-value. Dashed line denotes significance level of p<0.01.
DOI: https://doi.org/10.7554/eLife.34248.014
The following figure supplement is available for figure 3:

**Figure supplement 1.** Analyses of basal ganglia data using estimated action-values from the neurons' sessions.
DOI: https://doi.org/10.7554/eLife.34248.015

approximately 23% of the recorded neurons represent action-values at different phases of the experiment. As a first step, we estimated the action-values and regressed the spike counts in the different phases of the experiment on the estimated action-values, as in *Figure 1* (activity in each phase is analyzed as if it is a different neuron; see Materials and methods). The results of this analysis implied that 32% of the neurons represent action values (p<0.01) (*Figure 3—figure supplement 1*). Next, we applied the permutation analysis. Remarkably, this analysis yielded that only 3.6% of the neurons have a significantly higher regression coefficient on an action-value from their session than on other action-values (*Figure 3C*). Similar results were obtained when performing a similar model-free permutation analysis (regression of spike counts in the last 20 trials of the block on reward probabilities, not shown). These results raise the possibility that all or much of the apparent action-value representation in (*Ito and Doya, 2009*) is the result of the temporal correlations confound.

## Trial-design experiments

Another way of overcoming the temporal correlations confound is to use a trial design experiment. The idea is to randomly mix the reward probabilities, rather than use blocks as in *Figure 1*. For example, we propose the experimental design depicted in *Figure 4A*. Each trial is presented in one of four clearly marked contexts (color coded). The reward probabilities associated with the two actions are fixed within a context but differ between the contexts. Within each context the participant learns to prefer the action associated with a higher probability of reward. Naively, we can regress the spike counts on the action-values estimated from behavior, as in *Figure 1*. However, because the estimated action-values are temporally correlated, this regression is still subject to the temporal correlations confound (*Figure 2—figure supplement 9*). Alternatively, we can regress the spike counts on the reward probabilities. If the contexts are randomly mixed, then by construction, the reward probabilities are temporally independent. These reward probabilities are the objective action-values. After learning, the subjective action-values are expected to converge to these reward probabilities. Therefore, the reward probabilities can be used as proxies for the subjective action-values after a sufficiently large number of trials. It is thus possible to conduct a regression analysis on the spike counts at the end of the experiment, with reward probabilities as predictors that do not violate the independence assumption.

To demonstrate this method, we simulated learning in a session composed of 400 trials, randomly divided into 4 different contexts (*Figure 4*). Learning followed the Q-learning equations (*Equations 1 and 2*), independently for each context. Next, we simulated action-value neurons, whose firing rate is a linear function of the action-value in each trial (dots in *Figure 4A*, upper panel). We regressed the spike counts of the neurons in the last 200 trials (approximately 50 trials in each context) on the corresponding reward probabilities (*Figure 4B*). Indeed, 59% of the neurons were classified this way as action-value neurons (*Figure 4C*, 9.5% is chance level). By contrast, considering random-walk neurons, only 8.5% were erroneously classified as action-value neurons, a fraction expected by chance.

Three previous studies used trial-designs to search for action-value representation in the striatum (*Cai et al., 2011*; *FitzGerald et al., 2012*; *Kim et al., 2012*). In two of them (*Cai et al., 2011*; *Kim et al., 2012*) the reward probabilities were explicitly cued and therefore their results can be interpreted in the framework of cue-values and not action-values (*Padoa-Schioppa, 2011*). Moreover, all these studies focused on significant neural modulation by both action-values or by their difference, analyses that support state or policy representations (*Ito and Doya, 2015a*). As discussed in details in the next section, policy representation can emerge without action-value representation

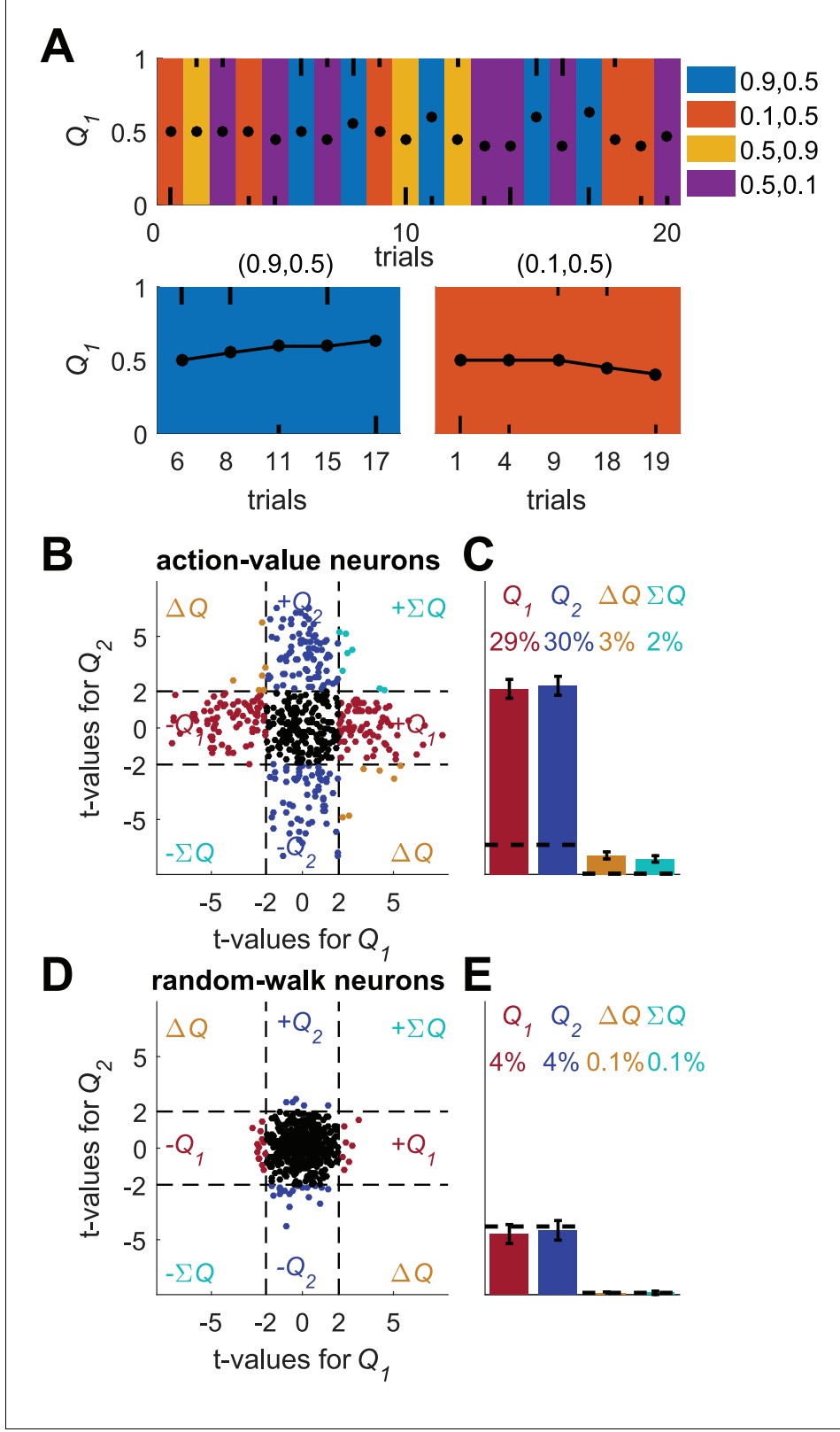

**Figure 4.** A possible solution for the temporal correlations confound that is based on a trial design. (**A**) A Q-learning model was simulated in 1,000 sessions of 400 trials, where the original reward probabilities (same as in *Figure 1A*) were associated with different cues and appeared randomly. Learning was done separately for each cue. Top panel: The first 20 trials in an example session. Background colors denote the reward probabilities in

*Figure 4 continued*

each trial. Black circles denote the learned value of action-value 1 in each trial. Top and bottom black lines denote choices of action 1 and 2, respectively. Long and short lines denote rewarded and unrewarded trials, respectively. Bottom panels: Two examples of the grouping of trials with the same reward probabilities to show the continuity in learning. Note that the action-value changes only when action 1 is chosen because it is the action-value associated with action 1. (**B**) and (**C**) population analysis for action-value neurons. 20,000 action-value neurons were simulated from the model in (**A**), similarly to the action-value neurons in *Figure 1*. For each neuron, the spike-counts in the last 200 trials of the session were regressed on the reward probabilities (see Materials and methods). Legend is the same as in *Figure 1D–E*. Dashed lines in (**B**) at t=2 denote the significance boundaries. Dashed lines in (**C**) denote the naïve expected false positive rate from the significance threshold (see Materials and methods). This analysis correctly identifies 59% of action-value neurons as such. (**D**) and (**E**) population analysis for random-walk neurons. 20,000 Random-walk neurons were simulated as in *Figure 2*. Same regression analysis as in (**B**) and (**C**). Only 8.5% of the random-walk neurons were erroneously classified as representing action-values (9.5% chance level).

DOI: https://doi.org/10.7554/eLife.34248.016

(*Darshan et al., 2014*; *Fiete et al., 2007*; *Frémaux et al., 2010*; *Loewenstein, 2008*; *Loewenstein, 2010*; *Loewenstein and Seung, 2006*; *Neiman and Loewenstein, 2013*; *Seung, 2003*; *Urbanczik and Senn, 2009*). Therefore, the results reported in (*Cai et al., 2011*; *FitzGerald et al., 2012*; *Kim et al., 2012*) cannot be taken as evidence for action-value representation in the striatum.

## Confound 2 – correlated decision variables

In the previous sections we demonstrated that irrelevant temporal correlations may lead to the erroneous classification of neurons as representing action-values, even if their activity is task-independent. Here we address an unrelated confound. We show that neurons that encode different decision variables, in particular policy, may be erroneously classified as representing action-values. For clarity, we will commence by discussing this caveat independently of the temporal correlations confound. Specifically, we show that neurons whose firing rate encodes the policy (probability of choice) may be erroneously classified as representing action-values, even when this policy emerged in the absence of any implicit or explicit action-value representation. We will conclude by discussing a possible solution that addresses this and the temporal correlations confounds.

### Policy without action-value representation

It is well-known that operant learning can occur in the absence of any value computation, for example, as a result of direct-policy learning (*Mongillo et al., 2014*). Several studies have shown that reward-modulated synaptic plasticity can implement direct-policy reinforcement learning (*Darshan et al., 2014*; *Fiete et al., 2007*; *Frémaux et al., 2010*; *Loewenstein, 2008*; *Loewenstein, 2010*; *Loewenstein and Seung, 2006*; *Neiman and Loewenstein, 2013*; *Seung, 2003*; *Urbanczik and Senn, 2009*).

For concreteness, we consider a particular reinforcement learning algorithm, in which the probability of choice $\Pr(a(t) = 1)$ is determined by a single variable $W$ that is learned in accordance with the REINFORCE learning algorithm (*Williams, 1992*): $\Pr(a(t) = 1) = \frac{1}{1+e^{-W(t)}}$ where $W(t) = \alpha \cdot (2 \cdot R(t) - 1) \cdot (a(t) - \Pr(a(t) = 1))$, where $\alpha$ is the learning rate, $R(t)$ is the binary reward in trial $t$ and $a(t)$ is a binary variable indicating whether action 1 was chosen in trial $t$. In our simulations $W(t = 1) = 0$, $\alpha = 0.17$. For biological implementation of this algorithm see (*Loewenstein, 2010*; *Seung, 2003*).

We tested this model in the experimental design of *Figure 1* (*Figure 5A*). As expected, the model learned to prefer the action associated with a higher probability of reward, completing the four blocks within 228 trials on average (standard deviation 62 trials).

### Spike count of neurons representing policy are correlated with estimated $\Delta Q$

Despite the fact that the learning was value-independent, we can still fit a Q-learning model to the behavior, extract best-fit model parameters and compute action-values (see also *Figure 2—figure*

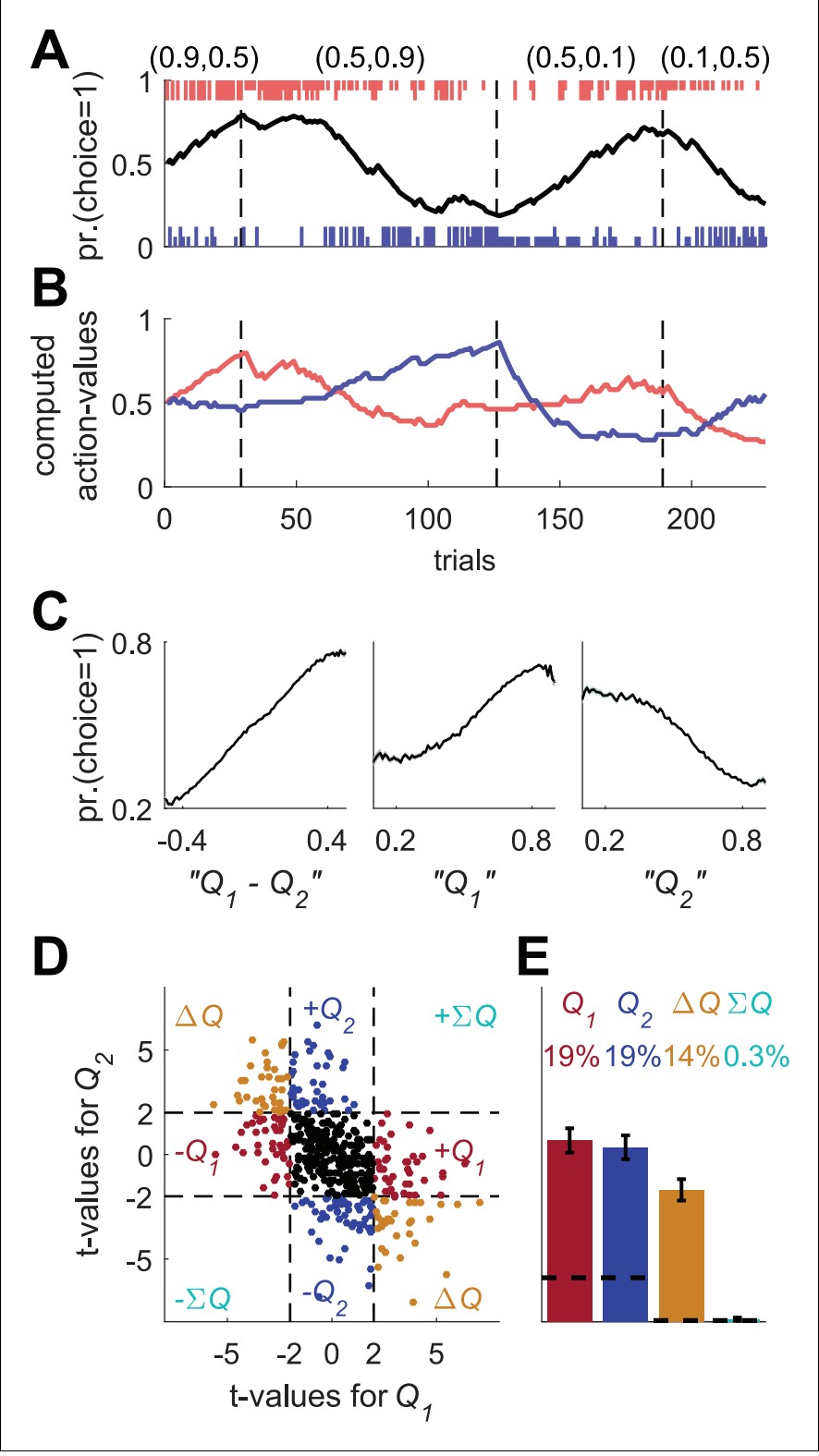

**Figure 5.** Erroneous detection of action-value representation in policy neurons. (**A**) Behavior of the model in an example session, same as in *Figure 1A* for the direct-policy model. (**B**) Red and blue lines denote 'action-values' 1 and 2, respectively, calculated from the choices and rewards in (**A**). Note that the model learned without any explicit or implicit calculation of action-values. The extraction of action-values in (**B**) is based on the fitting of *Equation 1* to the learning behavior. (**C**) Strong correlation between policy from the direct-policy algorithm and

*Figure 5 continued on next page*

*Figure 5 continued*

action-values extracted by fitting *Equation 1* to behavior. The three panels depict probability of choice as a function of the difference between the calculated action-values (left), '$Q_1$' (center) and '$Q_2$' (right). This correlation can cause policy neurons to be erroneously classified as representing action-values (D) and (E) Population analysis, same as in *Figure 1D and E* for the policy neurons. Legend and number of neurons are also as in *Figure 1D and E*. Dashed lines in (D) at t=2 denote the significance boundaries. Dashed lines in (E) denote the naïve expected false positive rate from the significance threshold (see Materials and methods).
DOI: https://doi.org/10.7554/eLife.34248.017

The following figure supplement is available for figure 5:

**Figure supplement 1.** Alternative analyses do not resolve the correlated decision variables confound.
DOI: https://doi.org/10.7554/eLife.34248.018

*supplement 1*). The computed action-values are presented in *Figure 5B*. Note that according to *Equation 2*, the probability of choice is a monotonic function of the difference between $Q_1$ and $Q_2$. Therefore, we expect that the probability of choice will be correlated with the computed $Q_1$ and $Q_2$, with opposite signs (*Figure 5C*).

We simulated policy neurons as Poisson neurons whose firing rate is a linear function of the policy $\Pr(a(t) = 1)$ (Materials and methods). Next, we regressed the spike counts of these neurons on the two action-values that were computed from behavior (same as in *Figures 1D,E* and *2B, C*, *Figure 2—figure supplement 1C,D*, – *Figure 2—figure supplement 2B,D*, – *Figure 2—figure supplement 3*). Indeed, as expected, 14% of the neurons were significantly correlated with both action values with opposite signs (chance level for each action value is 5%, naïve chance level for both with opposite signs is 0.125%, see Materials and methods), as depicted in *Figure 5D,E*. These results demonstrate that neurons representing value-independent policy can be erroneously classified as representing $\Delta Q$.

## Neurons representing policy may be erroneously classified as action-value neurons

Surprisingly, 38% of policy neurons were significantly correlated with *exactly* one estimated action-value, and therefore would have been classified as action-value neurons in the standard method of analysis (9.5% chance level).

To understand why this erroneous classification emerged, we note that a neuron is classified as representing an action-value if its spike count is significantly correlated with one of the action values, but not with the other. The confound that led to the classification of policy neurons as representing action-values is that *a lack of statistically significant correlation is erroneously taken to imply lack of correlation*. All policy neurons are modulated by the probability of choice, a variable that is correlated with the difference in the two action-values. Therefore, this probability of choice is expected to be correlated with both action-values, with opposite signs. However, because the neurons are Poisson, the spike count of the neurons is a noisy estimate of the probability of choice. As a result, in most cases (86%), the regression coefficients do not cross the significance threshold for *both* action-values. More often (38%), only one of them crosses the significance threshold, resulting in an erroneous classification of the neurons as representing action values.

## Is this confound relevant to the question of action-value representation in the striatum?

### If choice is included as a predictor, is policy representation still a relevant confound?

It is common, (although not ubiquitous) to attempt to differentiate action-value representation from choice representation by including choice as another regressor in the regression model (*Cai et al., 2011*; *FitzGerald et al., 2012*; *Funamizu et al., 2015*; *Her et al., 2016*; *Ito and Doya, 2015a*; *Ito and Doya, 2015b*; *Kim et al., 2013*; *Kim et al., 2009*; *Kim et al., 2012*; *Lau and Glimcher, 2008*). Such analyses may be expected to exclude policy neurons, whose firing rate is highly correlated with choice, from being classified as action-value neurons. However, repeating this analysis for

the policy neurons of *Figure 5*, we still erroneously classify 36% of policy neurons as action-value neurons (*Figure 5—figure supplement 1A*).

An alternative approach has been to consider only those neurons whose spike count is not significantly correlated with choice (*Stalnaker et al., 2010*; *Wunderlich et al., 2009*). Repeating this analysis for the *Figure 5* policy neurons, we still find that 24% of the neurons are erroneously classified as action-value neurons (8% are classified as policy neurons).

## Is this confound the result of an analysis that is biased against policy representation?

The analysis depicted in *Figures 1D,E*, *2B,C*, *4B–E* and *5D,E* is biased towards classifying neurons as action-value neurons, at the expense of state or policy neurons, as noted by (*Wang et al., 2013*). This is because action-value classification is based on a single significant regression coefficient whereas policy or state classification requires two significant regression coefficients. Therefore, (*Wang et al., 2013*) have proposed an alternative approach. First, compute the statistical significance of the whole regression model for each neuron (using f-value). Then, classify those significant neurons according to the t-values corresponding to the two action-values (*Figure 5—figure supplement 1B*). Applying this analysis to the policy neurons of *Figure 5* with a detection threshold of 5% we find that indeed, this method is useful in detecting which decision variables are more frequently represented (its major use in [*Wang et al., 2013*]): 25% of the neurons are classified as representing policy (1.25% expected by chance). Nevertheless, 12% of the neurons are still erroneously classified as action-value neurons (2.5% expected by chance; *Figure 5—figure supplement 1B*).

## Additional issues

In many cases, the term action-value was used, while the reported results were equally consistent with other decision variables. In some cases, significant correlation with both action-values (with opposite signs) or significant correlation with the difference between the action-values was used as evidence for 'action-value representations' (*FitzGerald et al., 2012*; *Guitart-Masip et al., 2012*; *Kim et al., 2012*; *2007*; *Stalnaker et al., 2010*). Similarly, other papers did not distinguish between neurons whose activity is significantly correlated with one action-value and those whose activity is correlated with both action-values (*Funamizu et al., 2015*; *Her et al., 2016*; *Kim et al., 2013*; *Kim et al., 2009*). Finally, one study used a concurrent variable-interval schedule, in which the magnitudes of rewards associated with each action were anti-correlated (*Lau and Glimcher, 2008*). In such a design, the two probabilities of reward depend on past choices and therefore, the objective values associated with the actions change on a trial-by-trial basis and are, in general, correlated.

## A possible solution to the policy confound

The policy confound emerged because policy and action-values are correlated. To distinguish between the two possible representations, we should seek a variable that is correlated with the action-value but uncorrelated with the policy. Consider the sum of the two action-values. It is easy to see that $\mathrm{Corr}(Q_1 + Q_2, Q_1 - Q_2) \propto \mathrm{Var}(Q_1) - \mathrm{Var}(Q_2)$. Therefore, if the variances of the two action-values are equal, their sum is uncorrelated with their difference. An action-value neuron is expected to be correlated with the sum of action-values. By contrast, a policy neuron, modulated by the difference in action-values is expected to be uncorrelated with this sum.

We repeated the simulations of *Figure 4* (which addresses the temporal correlations confound), considering three types of neurons: action-value neurons (of *Figure 1*), random-walk neurons (of *Figure 2*), and policy neurons (of *Figure 5*). As in *Figure 4*, we considered the spike counts of the three types of neurons in the last 200 trials of the session, but now we regressed them on the *sum* of reward probabilities (state; in this experimental design the reward probabilities are also the objective action-values, which the subject learns). We found that only 4.5 and 6% of the random-walk and policy neurons, respectively, were significantly correlated with the sum of reward probabilities (5% chance level). By contrast, 47% of the action-value neurons were significantly correlated with this sum.

This method is able to distinguish between policy and action-value representations. However, it will fail in the case of state representation because both state and action-values are correlated with the sum of probabilities of reward. To dissociate between state and action-value representations, we

can consider the difference in reward probabilities because this difference is correlated with the action-values but is uncorrelated with the state. Regressing the spike count on *both* the sum and difference of the probabilities of reward, a random-walk neuron is expected to be correlated with none, a policy neuron is expected to be correlated only with the difference, whereas an action-value neuron is expected to be correlated with both (this analysis is inspired by Fig. S8b in (*Wang et al., 2013*) in which the predictors in the regression model were policy and state). We now classify a neuron that passes both significance tests as an action-value neuron. Indeed, for a significance threshold of $p<0.05$ (for each test), only 0.2% of the random-walk neurons and 5% of the policy neurons were classified as action-value neurons. By contrast, 32% of the action-value neurons were classified as such (*Figure 6*). Note that in this analysis only when more than 5% of the neurons are classified as action-value neurons we have support for the hypothesis that there is action-value representation rather than policy or state representation.

A word of caution is that the analysis should be performed only after the learning converges. This is because stochastic fluctuations in the learning process may be reflected in the activities of neurons representing decision-related variables. As a result, policy or state-representing neurons may appear correlated with the orthogonal variables. For the same reason, any block-related heterogeneity in neural activity could also result in this confound (*O'Doherty, 2014*).

To conclude, it is worthwhile repeating the key features of the analysis proposed in this section:

1. Trial design is necessary because otherwise temporal correlations in spike count may inflate the fraction of neurons that pass the significance tests.
2. Regression should be performed on reward probabilities (i.e., the objective action-values) and not on estimated action-values. The reason is that because the estimated action-values evolve over time, this trial design does not eliminate all temporal correlations between them (*Figure 2—figure supplement 9*).
3. Reward probabilities associated with the two actions should be chosen such that their variances should be equal. Otherwise policy or state neurons may be erroneously classified as action-value neurons.

## Discussion

In this paper, we performed a systematic literature search to discern the methods that have been previously used to infer the representation of action-values in the striatum. We showed that none of these methods overcome two critical confounds: (1) neurons with temporal correlations in their firing rates may be erroneously classified as representing action-values and (2) neurons whose activity covaries with other decision variables, such as policy, may also be erroneously classified as representing action-values. Finally, we discuss possible experiments and analyses that can address the question of whether neurons encode action-values.

### Temporal correlations and action-value representations

It is well known in statistics that the regression coefficient between two independent slowly-changing variables is on average larger (in absolute value) than this coefficient when the series are devoid of a temporal structure. If these temporal correlations are overlooked, the probability of a false-positive is underestimated (*Granger and Newbold, 1974*). When searching for action-value representation in a block design, then by construction, there are positive correlations in the predictor (action-values). Positive temporal correlations in the dependent variable (neural activity) will result in an inflation of the false-positive observations, compared with the naïve expectation.

This confound occurs only when there are temporal correlations in both the predictor and the dependent variable. In a trial design, in which the predictor is chosen independently in each trial and thus has no temporal structure, we do not expect this confound. However, when studying incremental learning, it is difficult to randomize the predictor in each trial, making the task of identifying neural correlates of learning, and specifically action-values, challenging. With respect to the dependent variable (neural activity), temporal correlations in BOLD signal and their consequences have been discussed (*Arbabshirani et al., 2014*; *Woolrich et al., 2001*). Considering electrophysiological recordings, there have been attempts to remove these correlations, for example, using previous spike counts as predictors (*Kim et al., 2013*). However, these are not sufficient because they are

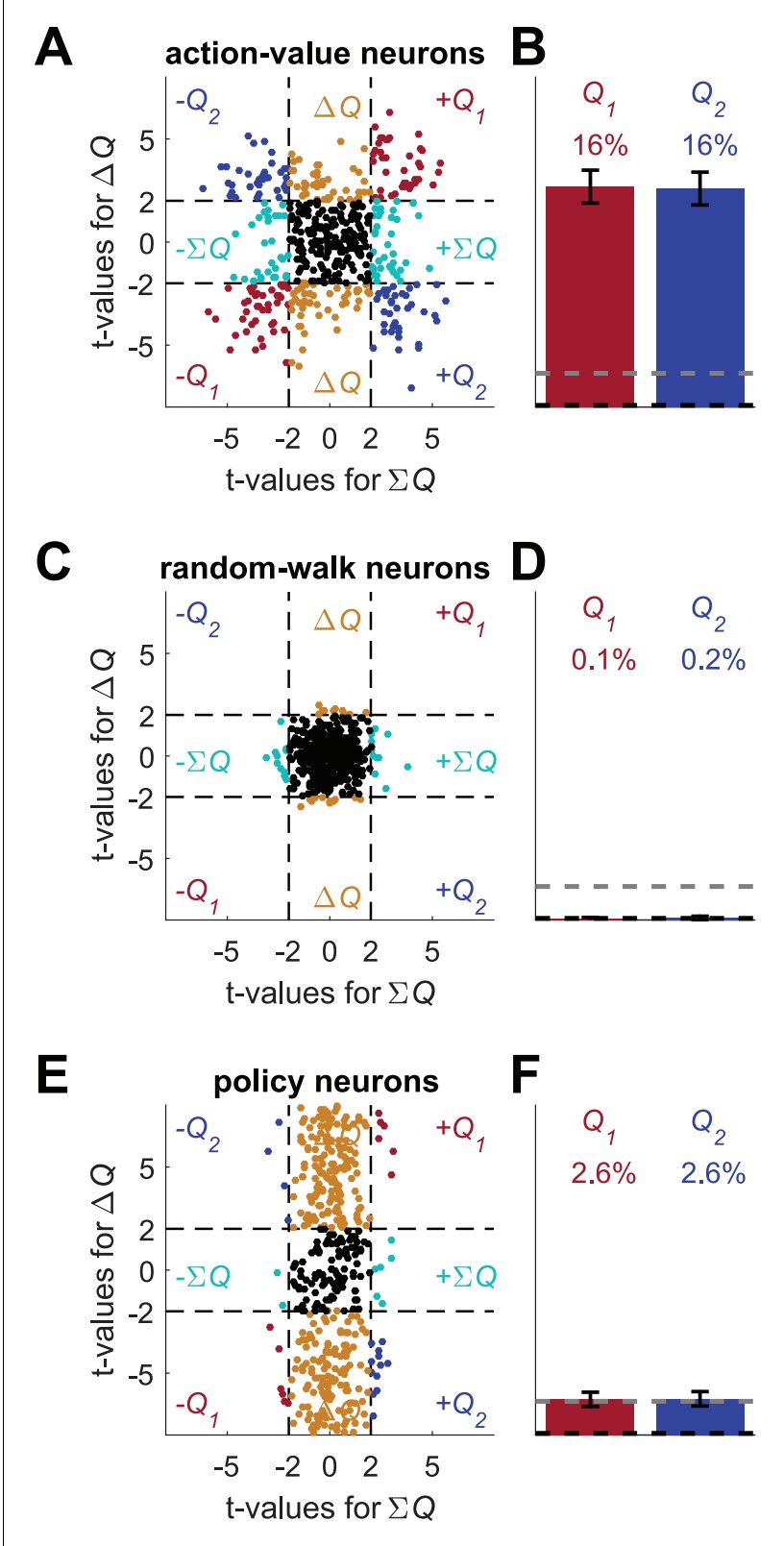

**Figure 6.** A possible solution for the policy and state confounds. (**A**) The Q-learning model (*Equations 1 and 2*) was simulated in 1,000 sessions of 400 trials each, where the reward probabilities were associated with different cues and were randomly chosen in each trial, as in *Figure 4*. Learning occurred separately for each cue. In each session 20 action-value neurons, whose firing rate is proportional to the action-values (as in *Figure 1*) were

*Figure 6 continued on next page*

*Figure 6 continued*

simulated. For each neuron, the spike-counts in the last 200 trials of each session were regressed on the sum of the reward probabilities ($\Sigma Q$; state) and the difference of the reward probabilities ($\Delta Q$; policy, see Materials and methods). Each dot denotes the t-values of the two regression coefficients of each of 500 example neurons. Dashed lines at t=2 denote the significance boundaries. Neurons that had significant regression coefficients on *both* policy and state were identified as action-value neurons. Colors as in *Figure 1D*. (B) Population analysis revealed that 32% of the action-value neurons were identified as such. Error bars are the standard error of the mean. Dashed black line denotes the expected false positive rate from randomly modulated neurons. Dashed gray line denotes the expected false positive rate from policy or state neurons (see Materials and methods) (C) Same as in (A) with random-walk neurons, numbers are as in *Figure 2*. (D) Population analysis revealed that less than 1% of the random-walk neurons were erroneously classified as representing action-values. (E-F) To test the policy neurons, we simulated a direct-policy learning algorithm (as in *Figure 5*) in the same sessions as in (A-D). Learning occurred separately for each cue. In each session 20 policy neurons, whose firing rate is proportional to the probability of choice (as in *Figure 5*) were simulated. As in (A-D), the spike-counts in the last 200 trials of each session were regressed on the sum and difference of the reward probabilities. (E) Each dot denotes the t-values of the two regression coefficients of each of 500 example neurons. (F) Population analysis. As expected, only 5% of the policy neurons were erroneously classified as representing action-values.

DOI: https://doi.org/10.7554/eLife.34248.019

unable to remove all task-independent temporal correlations (see also *Figure 2—figure supplements 4–10*). When repeating these analyses, we erroneously classified a fraction of neurons as representing action-value that is comparable to that reported in the striatum. The probability of a false-positive identification of a neuron as representing action-value depends on the magnitude and type of temporal correlations in the neural activity. Therefore, we cannot predict the fraction of erroneously classified neurons expected in various experimental settings and brain areas.

One may argue that the fact that action-value representations are reported mostly in a specific brain area, namely the striatum, is an indication that their identification there is not a result of the temporal correlations confound. However, because different brain regions are characterized by different spiking statistics, we expect different levels of erroneous identification of action-value neurons in different parts of the brain and in different experimental settings. Indeed, the fraction of erroneously identified action-value neurons differed between the auditory and motor cortices (compare B and D within *Figure 2—figure supplement 2*). Furthermore, many studies reported action-value representation outside of the striatum, in brain areas including the supplementary motor area and presupplementary eye fields (*Wunderlich et al., 2009*), the substantia nigra/ventral tegmental area (*Guitart-Masip et al., 2012*) and ventromedial prefrontal cortex, insula and thalamus (*FitzGerald et al., 2012*).

Considering the ventral striatum, our analysis on recordings from (*Ito and Doya, 2009*) indicates that the identification of action-value representations there may have been erroneous, resulting from temporally correlated firing rates (*Figure 3* and *Figure 2—figure supplement 3*). It should be noted that the fraction of action-value neurons reported in (*Ito and Doya, 2009*) is low relative to other publications, a difference that has been attributed to the location of the recording in the striatum (ventral as opposed to dorsal). It would be interesting to apply this method to other striatal recordings (*Ito and Doya, 2015a*; *Samejima et al., 2005*; *Wang et al., 2013*). We were unable to directly analyze these recordings from the dorsal striatum because relevant raw data is not publicly available. However, previous studies have reported that the firing rates of dorsal-striatal neurons change slowly over time (*Gouvêa et al., 2015*; *Mello et al., 2015*). As a result, identification of apparent action-value representation in dorsal-striatal neurons may also be the result of this confound.

Temporal correlations naturally emerge in experiments composed of multiple trials. Participants become satiated, bored, tired, etc., which may affect neuronal activity. In particular, learning in operant tasks is associated, by construction, with variables that are temporally correlated. If neural activity is correlated with performance (e.g., accumulated rewards in the last several trials) then it is expected to have temporal correlations, which may lead to an erroneous classification of the neurons as representing action-values.

## Temporal correlations – beyond action-value representation

Action-values are not the only example of slowly-changing variables. Any variable associated with incremental learning, motivation or satiation is expected to be temporally correlated. Even 'benign' behavioral variables, such as the location of the animal or the activation of different muscles may change at relatively long time-scales. When recording neural activity related to these variables, any temporal correlations in the neural recording, be it in fMRI, electrophysiology or calcium imaging may result in an erroneous identification of correlates of these behavioral variables because of the temporal correlation confound.

In general, the temporal correlation confound can be addressed by using the permutation analysis of *Figure 3*, which can provide strong support to the claim that the activity of a particular neuron or voxel co-varies with the behavioral variable. Therefore, the permutation test is a general solution for scientists studying slow processes such as learning. More challenging, however, is precisely identifying what the activity of the neuron represents (for example an action-value or policy). There are no easy solutions to this problem and therefore caution should be applied when interpreting the data.

## Differentiating action-value from other decision variables

Another difficulty in identifying action-value neurons is that they are correlated with other decision variables such as policy, state or chosen-value. Therefore, finding a neuron that is significantly correlated with an action-value could be the byproduct of its being modulated by other decision variables, in particular policy. The problem is exacerbated by the fact that standard analyses (e.g., *Figure 1D–E*) are biased towards classifying neurons as representing action-values at the expense of policy or state.).

As shown in *Figure 6*, policy representation can be ruled out by finding a representation that is orthogonal to policy, namely state representation. This solution leads us, however, to a serious conceptual issue. All analyses discussed so far are based on significance tests: we divide the space of hypothesis into the 'scientific claim' (e.g., neurons represent action-values) and the null hypothesis (e.g., neural activity is independent of the task). An observation that is not consistent with the null hypothesis is taken to support the alternative hypothesis.

The problem we faced with correlated variables is that the null hypothesis and the 'scientific claim' were not complementary. A neuron that represents policy is expected to be inconsistent with the null hypothesis that neural activity is independent of the task but it is not an action-value neuron. The solution proposed was to devise a statistical test that seeks to identify a representation that is correlated with action-value and is orthogonal to the policy hypothesis, in order to also rule out a policy representation.

However, this does not rule out other decision-related representations. A 'pure' action-value neuron is modulated only by $Q_1$ or by $Q_2$. A 'pure' policy neuron is modulated exactly by $Q_1 - Q_2$. More generally, we may want to consider the hypotheses that the neuron is modulated by a different combination of the action values, $a \cdot Q_1 + b \cdot Q_2$, where $a$ and $b$ are parameters. For every such set of parameters $a$ and $b$ we can devise a statistical test to reject this hypothesis by considering the direction that is orthogonal to the vector $(a, b)$. In principle, this procedure should be repeated for every pair of parameters $a$ and $b$ that in not consistent with the action-value hypothesis.

Put differently, in order to find neurons that represent action-values, we first need to *define* the set of parameters $a$ and $b$ such that a neuron whose activity is modulated by $a \cdot Q_1 + b \cdot Q_2$ will be considered as representing an action-value. Only after this (arbitrary) definition is given, can we construct a set of statistical tests that will rule out the competing hypotheses, namely will rule out *all* values of $a$ and $b$ that are not in this set. The analysis of *Figure 6* implicitly defined the set of $a$ and $b$ such that $a \neq b$ and $a \neq -b$ as the set of parameters that defines action-value representations. In practice, it is already very challenging to identify action-values using the procedure of *Figure 6* and going beyond it seems impractical. Therefore, studying the distribution of t-values across the population of neurons may be more useful when studying representations of decision variables than asking questions about the significance of individual neurons.

Importantly, the regression models described in this paper allow us to investigate only some types of representations, namely, linear combinations of the two action-values. However, value representations in learning models may fall outside of this regime. It has been suggested that in

decision making, subjects calculate the ratio of action-values (*Worthy et al., 2008*), or that subjects compute, for each action, the probability that it is associated with the highest value (*Morris et al., 2014*). Our proposed solution cannot support or refute these alternative hypotheses. If these are taken as additional alternative hypotheses, a neuron should be classified as representing an action-value if its activity is also significantly modulated in the directions that are correlated with action-value and are orthogonal to these hypotheses. Clearly, it is never possible to construct an analysis that can rule out all possible alternatives.

We believe that the confounds that we described have been overlooked because the null hypothesis in the significance tests was not made explicit. As a result, the complementary hypothesis was not explicitly described and the conclusions drawn from rejecting the null hypothesis were too specific. That is, alternative plausible interpretations were ignored. It is important, therefore, to keep the alternative hypotheses explicit when analyzing the data, be it using significance tests or other methods, such as model comparison (*Ito and Doya, 2015b*).

## Are action-value representations a necessary part of decision making?

One may argue that the question of whether neurons represent action-value, policy, state or some other correlated variable is not an interesting question. This is because all these correlated decision variables implicitly encode action-values. Even direct-policy models can be taken to implicitly encode action-values because policy is correlated with the difference between the action-values. However, we believe that the difference between action-value representation and representation of other variables is an important one, because it centers on the question of the computational model that underlies decision making in these tasks. Specifically, the implication of a finding that a population of neurons represents action-values is not that these neurons are involved somehow in decision making. Rather, we interpret this finding as supporting the hypothesis that action-values are explicitly computed in the brain, and that these action-values play a specific role in the decision making process. However, if the results are also consistent with various alternative computational models then this is not the case. Some consider action-value computation to be a necessary part of decision making. By contrast, however, we presented here two models of learning and decision making that do not entail this computation (*Figure 2—figure supplement 1*, *Figure 5*). Other examples are discussed in (*Mongillo et al., 2014*; *Shteingart and Loewenstein, 2014*) and references therein.

## Other indications for action-value representation

Several trial-design experiments have associated cues with upcoming rewards and reported representations of expected reward, the upcoming action, or the interaction of action and reward (*Cromwell and Schultz, 2003*; *Cromwell et al., 2005*; *Hassani et al., 2001*; *Hori et al., 2009*; *Kawagoe et al., 1998*; *Pasquereau et al., 2007*). Another trial-design experiment reported representation of offer-value and chosen-value in the orbitofrontal cortex (*Padoa-Schioppa and Assad, 2006*). While such studies do not provide direct evidence for action-value representation, they do provide evidence for representation of closely related decision variables (but see [*O'Doherty, 2014*]).

The involvement of the basal ganglia in general and the striatum in particular in operant learning, planning and decision-making is well documented (*Ding and Gold, 2010*; *McDonald and White, 1993*; *O'Doherty et al., 2004*; *Palminteri et al., 2012*; *Schultz, 2015*; *Tai et al., 2012*; *Thorn et al., 2010*; *Yarom and Cohen, 2011*). However, there are alternatives to the possibility that the firing rate of striatal neurons represents action-values. First, as discussed above, learning and decision making do not entail action-value representation. Second, it is possible that action-value is represented elsewhere in the brain. Finally, it is also possible that the striatum plays an essential role in learning, but that the representation of decision variables there is distributed and neural activity of single neurons could reflect a complex combination of value-related features, rather than 'pure' decision variables. Such complex representations are typically found in artificial neural networks (*Yamins and DiCarlo, 2016*).

## Action-value representation in the striatum requires further evidence

Considering the literature, both confounds have been partially acknowledged. Moreover, there have been some attempts to address them. However, as discussed above, even when these confounds

were acknowledged and solutions were proposed, these solutions do not prevent the erroneous identification of action-value representation (see *Figure 2—figure supplements 4*, *5* and *10*, *Figure 5—figure supplement 1*). We therefore conclude that to the best of our knowledge, all studies that have claimed to provide direct evidence that neuronal activity in the striatum is specifically modulated by action-value were either susceptible to the temporal correlations confound (*Funamizu et al., 2015*; *Ito and Doya, 2009*; *2015a*; *Ito and Doya, 2015b*; *Kim et al., 2013*; *Kim et al., 2009*; *Lau and Glimcher, 2008*; *Samejima et al., 2005*; *Wang et al., 2013*), or reported results in a manner indistinguishable from policy (*Cai et al., 2011*; *FitzGerald et al., 2012*; *Funamizu et al., 2015*; *Guitart-Masip et al., 2012*; *Her et al., 2016*; *Kim et al., 2013*; *Kim et al., 2009*; *Kim et al., 2012*; *2007*; *Stalnaker et al., 2010*; *Wunderlich et al., 2009*). Many studies presented action-value and policy representations separately, but were subject to the second confound (*Ito and Doya, 2009*; *2015a*; *Ito and Doya, 2015b*; *Lau and Glimcher, 2008*; *Samejima et al., 2005*). Furthermore, it should be noted that not all studies investigating the relation between striatal activity and action-value representation have reported positive results. Several studies have reported that striatal activity is more consistent with direct-policy learning than with action-value learning (*FitzGerald et al., 2014*; *Li and Daw, 2011*) and one noted that lesions to the dorsal striatum do not impair action-value learning (*Vo et al., 2014*).

Finally, we would like to emphasize that we do not claim that there is no representation of action-value in the striatum. Rather, our results show that special caution should be applied when relating neural activity to reinforcement-learning related variables. Therefore, the prevailing belief that neurons in the striatum represent action-values must await further tests that address the confounds discussed in this paper.

## Materials and methods

### Literature search

In order to thoroughly examine the finding of action-value neurons in the striatum, we conducted a literature search to find all the different approaches used to identify action-value representation in the striatum and see whether they are subject to at least one of the two confounds we described here.

The key words 'action-value' and 'striatum' were searched for in Web-of-Knowledge, Pubmed and Google Scholar, returning 43, 21 and 980 results, respectively. In the first screening stage, we excluded all publications that did not report new experimental results (e.g., reviews and theoretical papers), focused on other brain regions, or did not address value-representation or learning. In the remaining publications, the abstract of the publication was read and the body of the article was searched for 'action-value' and 'striatum'. After this step, articles in which it was possible to find description of action-value representation in the striatum were read thoroughly. The search included PhD theses, but none were found to report new relevant data, not found in papers. We identified 22 papers that directly related neural activity in the striatum to action-values. These papers included reports of single-unit recordings, fMRI experiments and manipulations of striatal activity.

Of these, two papers have used the term action-value to refer to the value of the *chosen* action (also known as chosen-value) (*Day et al., 2011*; *Seo et al., 2012*) and therefore we do not discuss them.

An additional study (*Pasquereau et al., 2007*) used the expected reward and the chosen action as predictors of the neuronal activity and found neurons that were modulated by the expected reward, the chosen action and their interaction. The authors did not claim that these neurons represent action-values, but it is possible that these neurons were modulated by the values of specific actions. However, the representation of the value of the action when the action is not chosen is a crucial part of action-value representation which differentiates it from the representation of expected reward, and the values of the actions when they were not chosen were not analyzed in this study. Therefore, the results of this study cannot be taken as an indication for action-value representation, rather than other decision variables.

A second group of 11 papers did not distinguish between action-value and policy representations (*Cai et al., 2011*; *Funamizu et al., 2015*; *Her et al., 2016*; *Kim et al., 2013*; *Kim et al., 2009*; *Wunderlich et al., 2009*), or reported policy representation (*FitzGerald et al., 2012*; *Guitart-*

*Masip et al., 2012*; *Kim et al., 2012*; *2007*; *Stalnaker et al., 2010*) in the striatum and therefore their findings do not necessarily imply action-value representation, rather than policy representation in the striatum (see confound 2).

In two additional papers, it was shown that the activation of striatal neurons changes animals' behavior, and the results were interpreted in the action-value framework (*Lee et al., 2015*; *Tai et al., 2012*). However, a change in policy does not entail an action-value representation (see, for example, *Figure 5* and *Figure 2—figure supplement 1*). Therefore, these papers were not taken as strong support to the striatal action-value representation hypothesis.

Finally, six papers correlated action-values, separately from other decision variables, with neuronal activity in the striatum (*Ito and Doya, 2009*; *2015a*; *Ito and Doya, 2015b*; *Lau and Glimcher, 2008*; *Samejima et al., 2005*; *Wang et al., 2013*). All of them used electrophysiological recordings of single units in the striatum. From these papers, only one utilized an analysis which is not biased towards identifying action-value neurons at the expense of policy and state neurons (*Wang et al., 2013*). All papers used block-design experiments where action-values are temporally correlated.

Taken together, we concluded that previous reports on action-value representation in the striatum could reflect the representation of other decision variables or temporal correlations in the spike count that are not related to action-value learning.

## The action-value neurons model (*Figure 1*, *Figure 4*)

To model neurons whose firing rate is modulated by an action-value, we considered neurons whose firing rate changes according to:

$$f(t) = B + K \cdot r \cdot (Q_i(t) - 0.5) \tag{4}$$

Where $f(t)$ is the firing rate in trial $t$, $B = 2.5$Hz is the baseline firing rate, $Q_i(t)$ is the action-value associated with one of the actions $i \in \{1, 2\}$, $K = 2.35$Hz is the maximal modulation and $r$ denotes the neuron-specific level of modulation, drawn from a uniform distribution, $r \sim U[-1, 1]$. The spike count in a trial was drawn from a Poisson distribution, assuming a 1 sec-long trial.

## The policy neurons model (*Figure 5*)

To model neurons whose firing rate is modulated by the policy, we considered neurons whose firing rate changes according to:

$$f(t) = B + K \cdot r \cdot (\Pr(a(t) = 1) - 0.5) \tag{5}$$

Where $f(t)$ is the firing rate in trial $t$, $B = 2.5$Hz is the baseline firing rate, $\Pr(a(t) = 1)$ is the probability of choosing action 1 in trial $t$ that changes in accordance with REINFORCE (*Williams, 1992*) (see also *Figure 5* and corresponding text). $K = 3$Hz is the maximal modulation and $r$ denotes the neuron-specific level of modulation, drawn from a uniform distribution, $r \sim U[-1, 1]$. The spike count in a trial was drawn from a Poisson distribution, assuming a 1 sec-long trial.

## The covariance neurons model (*Figure 2—figure supplement 1*)

In the covariance based plasticity model the decision-making network is composed of two populations of Poisson neurons: each neuron is characterized by its firing rate and the spike count of a neuron in a trial (1 sec) is randomly drawn from a Poisson distribution. The chosen action corresponds to the population that fires more spikes in a trial (*Loewenstein, 2010*; *Loewenstein and Seung, 2006*). At the end of the trial, the firing rate of each of the neurons (in the two population) is updated according to $f(t + 1) = f(t) + \eta \cdot R(t) \cdot (s(t) - f(t))$, where $f(t)$ is the firing rate in trial $t$, $\eta = 0.07$ is the learning rate, $R(t)$ is the reward delivered in trial $t$ ($R(t) \in \{0, 1\}$ in our simulations) and $s(t)$ is the measured (realized) firing rate in that trial, that is the spike count in the trial. The initial firing rate of all simulated neurons is 2.5Hz. The network model was tested in the operant learning task of *Figure 1*. A session was terminated (without further analysis) if the model was not able to choose the better option more than 14 out of 20 consecutive times for at least 200 trials in the same block. This occurred on 20% of the sessions. We simulated two populations of 1,000 neurons in 500 successful sessions. Note that because on average, the empirical firing rate is equal to the true firing rate, $f(t) = \langle s(t) \rangle$, changes in the firing rate are driven, on average, by the covariance of reward and the empirical firing rate: $\langle f(t) \rangle \equiv \langle f(t+1) - f(t) \rangle = \eta \cdot \text{cov}(R(t), s(t))$ (*Loewenstein and Seung, 2006*).

The estimated action-values in *Figure 2—figure supplement 1* were computed from the actions and rewards of the covariance model by assuming the Q-learning model (*Equations 1 and 2*).

## The motor cortex recordings (*Figure 2—figure supplement 2*)

The data in *Figure 2—figure supplement 2A–B* was recorded by Oren Peles in Eilon Vaadia's lab. It was recorded from one female monkey (Macaca fascicularis) at 3 years of age, using a $10 \times 10$ microelectrode array (Blackrock Microsystems) with 0.4 mm inter-electrode distance. The array was implanted in the arm area of M1, under anesthesia and aseptic conditions.

Behavioral Task: The Monkey sat in a behavioral setup, awake and performing a Brain Machine Interface (BMI) and sensorimotor combined task. Spikes and Local Field Potentials were extracted from the raw signals of 96 electrodes. The BMI was provided through real time communication between the data acquisition system and a custom-made software, which obtained the neural data, analyzed it and provided the monkey with the desired visual and auditory feedback, as well as the food reward. Each trial began with a visual cue, instructing the monkey to make a small hand movement to express alertness. The monkey was conditioned to enhance the power of beta band frequencies (20-30Hz) extracted from the LFP signal of 2 electrodes, receiving a visual feedback from the BMI algorithm. When a required threshold was reached, the monkey received one of 2 visual cues and following a delay period, had to report which of the cues it saw by pressing one of two buttons. Food reward and auditory feedback were delivered based on correctness of report. The duration of a trial was on average 14.2s. The inter-trial-interval was 3s following a correct trial and 5s after error trials. The data used in this paper, consists of spiking activity of 89 neurons recorded during the last second of inter-trial-intervals, taken from 600 consecutive trials in one recording session. Pairwise correlations were comparable to previously reported (*Cohen and Kohn, 2011*), $r_{SC} = 0.047 \pm 0.17$ (SD), ($r_{SC} = 0.037 \pm 0.21$ for pairs of neurons recorded from the same electrode).

Animal care and surgical procedures complied with the National Institutes of Health Guide for the Care and Use of Laboratory Animals and with guidelines defined by the Institutional Committee for Animal Care and Use at the Hebrew University.

## The auditory cortex recordings (*Figure 2—figure supplement 2*)

The auditory cortex recordings appearing in *Figure 2—figure supplement 2C–D* are described in detail in (*Hershenhoren et al., 2014*). In short, membrane potential was recorded intracellularly in the auditory cortex of halothane-anesthetized rats. The data consists of 125 experimental sessions recorded from 39 neurons. Each session consisted of 370 pure tone bursts. Tone duration was 50 ms with 5 ms linear rise/fall ramps. In the data presented here, trials began 50 ms prior to the onset of the tone burst. For each session, all trials were either 300 msec or 500 msec long. Trial length remained identical throughout a session and depended on smallest interval between two tones in each session. Spike events were identified following high pass filtering with a corner frequency of 30Hz. Local maxima that were larger than 60 times the median of the absolute deviation from the median (MAD) were classified as spikes. The data presented here consists only of the spike counts in each trial, rather than the full membrane potential trace.

## The basal ganglia recordings (*Figure 3* and *Figure 2—figure supplement 3*)

The basal ganglia recordings that are analyzed in *Figure 3* and *Figure 2—figure supplement 3* are described in detail in (*Ito and Doya, 2009*). In short, rats performed a combination of a tone discrimination task and a reward-based free-choice task. Extracellular voltage was recorded in the behaving rats from the NAc and VP using an electrode bundle. Spike sorting was done using principal component analysis. In total, 148 NAc and 66 VP neurons across 52 sessions were used for analyses (In 18 of the 70 behavioral sessions there were no neural recordings).

## Estimation of action-values from choices and rewards

To imitate experimental procedures, we regressed the spike counts on estimates of the action-values, rather than the subjective action-values that underlay model behavior (to which the experimentalist has no direct access). For that goal, for each session, we assumed that $Q_i(1) = 0.5$ and found the set of parameters $\alpha$ and $\beta$ that yielded the estimated action-values that best fit the sequences of

actions in each experiment by maximizing the likelihood of the sequence. Action-values were estimated from *Equation 1*, using these estimated parameters and the sequence of actions and rewards. Overall, the estimated values of the parameters $\alpha$ and $\beta$ were comparable to the actual values used: on average, $\alpha = 0.12 \pm 0.09$ (standard deviation) and $\beta = 2.6 \pm 0.7$ (compare with $\alpha=0.1$ and $\beta=2.5$).

## Exclusion of neurons

Following standard procedures (*Samejima et al., 2005*), a sequence of spike-counts, either simulated or experimentally measured was excluded due to low firing rate if the mean spike count in all blocks was smaller than 1. This procedure excluded 0.02% (4/20,000) of the random-walk neurons and 0.03% (285/1,000,000) of the covariance-based plasticity neurons. Considering the auditory cortex recordings, we assigned each of the 125 spike counts to 40 randomly-selected sessions. 23% of the neural recordings (29/125) were excluded in all 40 sessions. Because blocks are defined differently in different sessions, some neural recordings were excluded only when assigned to some sessions but not others. Of the remaining 96 recordings, 14% of the recordings × sessions were also excluded. Similarly, considering the basal ganglia neurons, we assigned each of the 642 recordings (214 × 3 phases) to 40 randomly-selected sessions. 11% (74/(214 × 3)) of the recordings were excluded in all 40 sessions. Of the remaining 568 recordings, 9% of the recordings × sessions were also excluded. None of the simulated action-value neurons (0/20,000) or the motor cortex neurons (0/89) were excluded.

## Statistical analyses

The computation of the t-values of the regression of the spike counts on the estimated action-values (as in *Figures 1*, *2* and *5*, *Figure 2—figure supplement 1*, – *Figure 2—figure supplement 2*, –*Figure 2—figure supplement 3*) was done using the following regression model:

$$s(t) = \beta_0 + \beta_1 Q_1(t) + \beta_2 Q_2(t) + \epsilon(t) \tag{6}$$

Where $s(t)$ is the spike count in trial $t$, $Q_1(t)$ and $Q_2(t)$ are the estimated action-values in trial $t$, $\epsilon(t)$ is the residual error in trial $t$ and $\beta_{0-2}$ are the regression parameters.

The computation of the t-values of the regression of the spike counts on the reward probabilities in the trial design experiment (as in *Figure 4*) was done using the following regression model:

$$s(t) = \beta_0 + \beta_1 RP_1(t) + \beta_2 RP_2(t) + \epsilon(t) \tag{7}$$

Where $t$ denotes the trial. Only the last 200 trials of the session were anlyzed. $s(t)$ is the mean spike count, $RP_1(t)$ and $RP_2(t)$ are the reward probabilities corresponding to action 1 or action 2, respectively (in this experimental design $RP$ could be 0.1,0.5 or 0.9), $\epsilon(t)$ is the residual error and $\beta_{0-2}$ are the regression parameters.

The computation of the t-values of the regression of the spike counts on *state* and *policy* in a trial design experiment (as in *Figure 6*) was done using the following regression model:

$$s(t) = \beta_0 + \beta_1 [RP_1(t) + RP_2(t)] + \beta_2 [RP_1(t) - RP_2(t)] + \epsilon(t) \tag{8}$$

All variables and parameters are the same as in *Equation 7*

All regression analyses were done using *regstats* in MATLAB (version 2016A).

To compare the spike counts of the example neurons, in the last 20 trials of each block (*Figure 1B*; *Figure 2—figure supplement 1B*; *Figure 2—figure supplement 2A*; *Figure 2—figure supplement 2C*; *Figure 2A*) we executed the Wilcoxon rank sum test, using *ranksum* in MATLAB. All tests were two-tailed.

Significance of t-values slightly depends on session length. For the session lengths we considered, 0.05 significance bounds varied between 1.962 and 1.991. For consistency, we chose a single conservative bound of 2. Similarly, 0.025 and 0.01 significance bounds were chosen to be 2.3 and 2.64, respectively.

For all significance boundaries the false positive thresholds were computed naively, that is, assuming the analysis is not confounded in any way and that the two predictors are not correlated with each other. For example, assuming the false positive rate from a single t-test for a significant

regression coefficient is $P$, for the standard analysis, the false positive rate for each action-value classification was defined as $P \cdot (1 - P)$, and the false positive rate was equal for state and policy classification and was defined as $P^2/2$. In *Figure 6* the false positive rate computed for random-walk neurons was $P^2/2$ for each action-value classification, and the false positive rate computed for state or policy neurons was $P/2$ for each action-value classification.

### Permutation test (*Figure 3*)

For each action-value and random-walk neuron, we computed the t-values of the regressions of its spike-count on estimated action-values from the sessions of *Figure 1E*. Because the number of trials can affect the distribution of t-values, we only considered in our analysis the first 170 trials of the 504 sessions longer or equal to 170 trials. This number, which is approximately the median of the distribution of number of trials per session, was chosen as a compromise between the number of trials per session and number of sessions. When performing the permutation test on the basal ganglia data we included all recordings and only the first 332 trials in each session, which is the smallest number of trials used in a session in this dataset.

Two points are noteworthy. First, the distribution of the t-values of the regression of the spike count of a neuron on all action-values depends on the neuron (see difference between distributions in *Figure 3A*). Similarly, the distribution of the t-values of the regression of the spike counts of all neurons on an action-value depends on the action-value (not shown). Therefore, the analysis could be biased in favor (or against) finding action-value neurons if the number of neurons analyzed from each session (and therefore are associated with the same action-values) differs between sessions. Second, this analysis does not address the correlated decision variables confound.

Finally, we would like to point out that there is an alternative way of performing the permutation test, which is applicable when the number of sessions is small, while the number of neurons recorded in a session is large. Instead of comparing the t-values from the regression of a neuron on different action-values, one can compare the t-values from different neurons on the same action-value. However, this method is only applicable under the assumption that the temporal correlations that are not related to action-value in the neuronal activity are similar between sessions.

### Comparison with permuted spike counts (*Figure 2—figure supplement 4*)

In *Figure 2—figure supplement 4* we considered the experiment and analysis described in (*Kim et al., 2009*). That experiment consisted of four blocks, each associated with a different pair of reward probabilities, (0.72, 0.12), (0.12, 0.72), (0.21, 0.63) and (0.63, 0.21), appearing in a random order, with the better option changing location with each block change. The number of trials in a block was preset, ranging between 35 and 45 with a mean of 40 (this is unlike the experiment described in *Figure 1*, in which termination of a block depended on performance).

First, we used *Equations 1 and 2* to model learning behavior in this protocol. Then, we estimated the action-values according to choice and reward sequences, as in *Figure 1*. These estimated action-values were used for regression of the spike counts of the random-walk, motor cortex, auditory cortex, and basal ganglia neurons in the following way: each spike count sequence was randomly assigned to a particular pair of estimated action-values from one session. The spike count sequence was regressed on these estimated action-values. The resultant t-values were compared with the t-values of 1000 regressions of the spike-count, permuted within each block, on the same action-values. The p-value of this analysis was computed as the percentage of t-values from the permuted spike-counts that were higher in absolute value than the t-value from the regression of the original spike count. The significance boundary was set at p<0.025 (*Kim et al., 2009*). Neurons with at least one significant regression coefficient (rather than exactly one significant regression coefficient) were classified as action-value modulated neurons (*Kim et al., 2009*).

### ANOVA tests for comparisons between blocks, excluding 'drifting' neurons

Following (*Asaad et al., 2000*) we conducted an additional analysis with repeating blocks. We simulated learning behavior in the same experiment as in *Figure 2—figure supplement 10*. This experiment is composed of 8 blocks - the 4 blocks of *Figure 1*, repeated twice, in random permutation.

We restricted our analysis to the 438 sessions with 332 trials or fewer (332 trials is the shortest session in the basal ganglia recording). Each spike count was analyzed 40 times, using 40 randomly-assigned sessions. For each block, we restricted the analysis to the neuronal activity in the last 20 trials of the block.

First, we conducted four one-way ANOVAs (using MATLAB's anova1) to compare the neuronal activities in blocks associated with the same action-values (e.g., the neuronal activity in the two blocks, in which reward probabilities were (0.1,0.5)). Neurons were excluded from further analysis if we found a significant difference in their firing rates in at least one of these comparisons (df(columns)=1, df(error)=38, p<0.1). This procedure excludes from further analysis 'drifting' neurons, whose spike count significantly varied in the session.

Next, for each action-value we conducted a one-way ANOVA (using MATLAB's anova1), which compared the neuronal activity between the two blocks in which the action-value was 0.1 and the two blocks in which the action-value was 0.9 (df(columns)=1, df(error)=78, p<0.01). We classified neurons as representing action-values if there was a significant difference between their firing rates for one action-value but not for the other.

Despite the removal of 'drifting' neurons, this analysis yielded an erroneous classification of action-value neurons in all datasets: random-walk neurons, 18%; motor cortex neurons, 12%; auditory cortex neurons, 5%; basal ganglia neurons, 9%. This is despite the fact that the expected false positive rate is only 2%. These results indicate that the exclusion of 'drifting' neurons as in (*Asaad et al., 2000*) does not solve the temporal correlations confound.

Data from the motor cortex, auditory cortex, and basal ganglia was the same as in *Figure 2—figure supplements 2–3*. Data for random-walk included 1000 newly simulated neurons, using the same parameters as in *Figure 2* (this was done to create enough trials in each simulated spike count).

## Data and code availability

The data of the basal ganglia recordings from (*Ito and Doya, 2009*) is available online at https://groups.oist.jp/ncu/data and was analyzed with permission from the authors. Motor cortex data (recorded by Oren Peles in Eilon Vaadia's lab) and auditory cortex data (taken from the recordings in (*Hershenhoren et al., 2014*)) is available at https://github.com/lotem-elber/striatal-action-value-neurons-reconsidered-codes (*Elber-Dorozko and Loewenstein, 2018*). The custom MATLAB scripts used to create simulated neurons and to analyze simulated and recorded neurons are also available at https://github.com/lotem-elber/striatal-action-value-neurons-reconsidered-codes (*Elber-Dorozko and Loewenstein, 2018*; copy archived at https://github.com/elifesciences-publications/striatal-action-value-neurons-reconsidered-codes).

## Acknowledgements

We are extremely grateful to Oren Peles, Eilon Vaadia and Uri Werner-Reiss for providing us with their motor cortex recordings, Bshara Awwad, Itai Hershenhoren, Israel Nelken for providing us with their auditory cortex recordings, Kenji Doya and Makoto Ito for providing us with their basal ganglia recordings, Mati Joshua, Gianluigi Mongillo, Jonathan Roiser and Roey Schurr for careful reading of the manuscript and helpful comments and Inbal Goshen, Hanan Shteingart and Wolfram Schultz for discussions.

## Additional information

### Funding

| Funder | Grant reference number | Author |
| --- | --- | --- |
| Israel Science Foundation | 757/16 | Yonatan Loewenstein |
| Deutsche Forschungsge-meinschaft | CRC1080 | Yonatan Loewenstein |
| Gatsby Charitable Foundation | | Yonatan Loewenstein |

The funders had no role in study design, data collection and interpretation, or the decision to submit the work for publication.

## Author contributions
Lotem Elber-Dorozko, Conceptualization, Formal analysis, Visualization, Methodology, Writing—original draft, Writing—review and editing; Yonatan Loewenstein, Conceptualization, Formal analysis, Supervision, Visualization, Methodology, Writing—original draft, Writing—review and editing

## Author ORCIDs
Lotem Elber-Dorozko (iD) http://orcid.org/0000-0003-1235-8651
Yonatan Loewenstein (iD) http://orcid.org/0000-0003-2577-2317

## Decision letter and Author response
Decision letter https://doi.org/10.7554/eLife.34248.028
Author response https://doi.org/10.7554/eLife.34248.029

# Additional files

## Supplementary files
• Transparent reporting form
DOI: https://doi.org/10.7554/eLife.34248.020

## Data availability
The data of the basal ganglia recordings from (Ito and Doya 2009) is available online at https://groups.oist.jp/ncu/data and was analyzed with permission from the authors. Motor cortex data (recorded by Oren Peles in Eilon Vaadia's lab) and auditory cortex data (taken from the recordings in (Hershenhoren, Taaseh, Antunes, & Nelken, 2014)) is available at https://github.com/lotem-elber/striatal-action-value-neurons-reconsidered-codes (Elber-Dorozko & Loewenstein 2018). The custom MATLAB scripts used to create simulated neurons and to analyze simulated and recorded neurons are also available at https://github.com/lotem-elber/striatal-action-value-neurons-reconsidered-codes (copy archived at https://github.com/elifesciences-publications/striatal-action-value-neurons-reconsidered-codes).

The following previously published datasets were used:

| Author(s) | Year | Dataset title | Dataset URL | Database, license, and accessibility information |
| --- | --- | --- | --- | --- |
| Ito M, Doya K | 2009 | Validation of decision making models and analysis of decision variables in the rat basal ganglia. | https://groups.oist.jp/ncu/data | Publicly available at OIST Groups website. |

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
