## [Decision Letter]

Thank you for submitting your article "Striatal action-value neurons reconsidered" for consideration by *eLife*. Your article has been reviewed by three peer reviewers, and the evaluation has been overseen by a Reviewing Editor and Timothy Behrens as the Senior Editor. The reviewers have opted to remain anonymous.

The reviewers have discussed the reviews with one another and the Reviewing Editor has drafted this decision to help you prepare a revised submission.

Summary:

Elber-Dorozko and Loewenstein examine issues using trial-by-trial spike data to determine whether neural activity is associated with action-values. First, they note that standard regression inference can lead to false detection of action-value correlations when samples are not independent. They illustrate the prevalence of false detections using simulation and data with no plausible relation to action-values. Second, they note that different reinforcement learning models without action-value representations can yield significant action-value coding when analyzed using prior approaches. The authors present well-thought-out analyses and data that highlight analytic, experimental and conceptual difficulties in identifying action-value coding. Although both issues the authors examine have been acknowledged in the literature, and different approaches have been made to deal with or minimize their potential effects, the authors' paper represents an important synthesis and analysis that will likely lead to clearer future experiments.

However, there were several key concerns.

Firstly, the reviewers thought that there should be a more careful attention to the wider literature. In the reviewer discussion, this was thought to be of particular importance as you are making a technical point in a journal with a broad readership. It is particularly important that you are clear about which parts of the literature your results speak to.

For example:

1) The authors conclude that current methods of studying action-values are confounded, and propose that experiments should use actions whose reward values are not learned over time during the experiment, and instead are indicated by sensory cues with their values picked randomly on each trial.

However, the authors leave unmentioned the large literature of studies taking an alternate approach: recording striatal activity during planning and execution of instructed actions for cued reward outcomes, in which each trial randomizes the instructed action, the reward, or both. The authors should discuss what implications this literature has for whether the striatum encodes action-values vs. policies vs. other variables. Examples include the work from the labs of Schultz (e.g. Hassani et al., 2001; Cromwell and Schultz, 2003; Cromwell et al., 2005), Hikosaka (e.g. Kawagoe et al., 1998; Lauwereyns et al., Neuron 2004), and Kimura (e.g. Hori et al., 2009).

Most of these studies did not specifically claim that the activity they reported represents "action-values" in the reinforcement learning sense (and hence the present authors shouldn't feel obligated to try to debunk them), but they do seem highly relevant to the larger question the present authors raise. These studies did attempt to test whether neurons represented actions, values, and notably, their interaction (e.g. a cell whose activity scales more with action A's value than action B's value), which resembles the concept of "action-value".

Also, these studies may be somewhat resistant to the authors' criticisms about confounds from temporal correlations (since rewards were either explicitly cued, or kept deterministic and stable in well-learned blocks of trials, rather than slowly fluctuating during extended learning) and confounds with action probability (since the actions were instructed and hence a priori equally probable on each trial).

Of particular interest, a paper by Pasquereau et al. (2007) seems to fulfill all the requirements the present authors set for a test of striatal action-value coding; if so, this seems worthy of mention. That study manipulated the reward value of four actions (up, down, left, right), randomly assigning their reward probabilities on each trial and indicating them with visual cues. Unfortunately, as the present authors note, the study did not explicitly analyze their results in the action-value vs. chosen-value framework. However, the paper did report that some neurons had significant action x value interactions – for example, a cell that is more active when planning rightward movements (action), with stronger activity when the planned movement was more valuable (value), and with this value-modulation greater for rightward movements than other movements (action x value). This is not a pure chosen-value signal as the present authors seem to claim that paper reported. One could argue that it contains a key feature of an action-value signal as the value modulation is strongest for one specific action.

2) The authors correctly pointed out that some earlier studies of action value used a suboptimal task design and their conclusions need to be more rigorously validated. However, in the broader field, the potential risk of "drift" in neural recording has been well recognized. For example, "Neurophysiological experiments that compare activity across different blocks of trials must make efforts to be confident that any neural effects are not the result of artifacts of that design, such as slow-wave changes in neural activity over time." (Asaad et al., 2000). In the same Asaad et al. paper, a better design with repeated, alternated block types was used, similar in concept to randomized block design that the authors proposed here. Such designs have also been used in many neural studies of cognition – to name just two examples: value manipulations (Lauwereyns et al., 2002), rule manipulations (Mansouri et al., 2006). The problem thus seems relatively limited to one type of analysis that introduces temporal correlation across trials in an effort to estimate Q values. By the authors' account, this amounts to 5 papers from 3 different labs.

3) What about previous results arguing for prominence of a specific type of value representation? The authors touch on this, but it would be helpful to discuss specific results. In particular, the cited study of Wang et al., Nat Neuro 2003 reported that their unbiased angular measure of DMS value coding was distributed significantly non-uniformly, with net value (Σ*Q*) coding more prevalent than other types (their Figure 7). Whereas the null hypothesis simulations in this paper predict very different results, either a uniform distribution (Figure 2—figure supplement 8) or a dearth of Σ*Q* neurons (Figure 5—figure supplement 1). The authors should discuss whether this previous result can therefore still be interpreted as evidence of value coding (at least, net-value coding), rather than strictly policy coding, in the striatum.

(Also, it is odd that the authors cite Wang et al. as a study that "claimed to provide direct evidence that neuronal activity in the striatum is specifically modulated by action-value", since the main result was specifically finding prevalence of net-value, *not* action-value coding).

4) I found the authors' choice of basal ganglia data misleading (Ito and Doya, 2009). First, because these data are recordings from the nucleus accumbens and ventral pallidum, which are not the first basal ganglia structures one thinks of as encoding action values. Second, because the original authors of the study from which the data was obtained noted that action value coding was low in these structures, leading them to suggest that action value coding was not the primary function of the nucleus accumbens and ventral pallidum. This is mentioned in the Results subsection “Permutation analysis of basal ganglia neurons”, but should be noted in the Discussion (the current text in the Results could probably just be moved).

5) Previous methods dealing with trial correlations have different success at reducing false positive rate of detecting action-values. In particular, the method of Wang et al. (2013) comes very close to attaining the correct size of the test for action-values. Indeed, it appears to be the only existing method from which one would reasonably conclude that the ventral striatal data set analyzed probably does not exhibit much action-value coding (2-3% above the expected 5%). I think it would be useful to have a figure in the main text comparing the different methods to the authors' permutation test (using for example, just the basal ganglia data set). In addition, Wang et al.’s method is also pretty good at identifying policy neurons, which is important because it could be applied retrospectively to existing data sets.

6) The authors' biggest suggestion for rigorously detecting a neural action value representation is "Don't use a task with learning, use a trial-based design where subjects associate contexts with well-learned sets of action values". That is perfectly fine for scientists whose goal is specifically to test whether a brain area encodes action values. However, what about the many scientists whose explicit goal is to study neural representation of time-varying values, and hence *need* to use learning tasks? Many scientists are studying (1) the neural basis of value learning, (2) brain areas specifically involved in early learning (not well-trained performance), (3) motivational variables specifically present for time-varying action values not well-trained ones (e.g. volatility, certain forms of uncertainty, etc.). If the authors can give an approach that will let these scientists make accurate estimates of neural time-varying value coding during learning tasks, that would certainly be valuable to the field.

I feel like their methods could potentially be used to achieve this in a straightforward way (by combining their novel permutation test from Part 1 of their paper with their method of testing for correlation with both sum and difference of values from Part 2 of their paper). But they don't lay this out explicitly in their paper at present, since they are more focused on the narrower implication ("Do striatal neurons encode action values?") rather than the broader implication of their results ("In general, how can one properly measure time-varying action values?").

Secondly, the reviewers had some particular concerns about the action value vs. policy representation issue. For example:

1) Regarding the second confound of policy vs. action value:

- The authors seem to be arguing against a straw-man version of how to relate neural activity to behavior. Typically we infer the underlying computations by testing how well different hypothesized models can fit the behavior and then search for correlates of the most likely computation in the brain. The authors seem to test only how well the neural activity correlates with different hypothesized models.

- The proposed solution for distinguishing policy from action value also has a very high rate of false negative (78%).

2) I feel that the point the authors make about action-value vs. policy representations may actually be underselling the true extent of the confound, and so their proposed solution may not be sufficient. However, this all depends on how the authors want to define a 'policy neuron' vs. a 'value neuron', as I explain below. I think they should clarify this.

2.1) Their arguments seem to assume that neural policy representations are in the form of action probabilities, which can then be identified by the key signature that they relate to action-values in a 'relative' manner (e.g. an 'action 1' neuron that is correlated positively with *Q*_1_ must be correlated negatively with *Q*_2_), and hence will be best fit as encoding Δ*Q (Q*_1_ – *Q*_2_). However, depending on how they define 'policy', this may not be the case.

Notably, even for reinforcement learning agents that do not explicitly represent action-values, few of them directly learn a policy in its most raw form of a set of action probabilities. Instead, they represent the policy in a more abstract parameter space. The simplest parameterization is a vector of action strengths, one for each possible action. Then during a choice, the probability of each action is determined by applying a choice function (e.g. softmax) to the action strengths of the set of actions that are currently available. The choice's outcome is then used to do learning on the action strengths. This method is used by some traditional actor-critic agents (which represent state values and action strengths, but not action-values). My impression is that the authors' covariance-based model is similar, in that the variables that it updates when it learns are the input weights *W*_1_ and *W*_2_ to each pool (with one input weight for each action, thus being analogous to action strengths).

Note that in these models, the action *strengths* are not explicitly represented in a 'relative' manner; only the resulting action *probabilities* are (since the probabilities must sum to 1). It's not clear to me whether a neuron encoding an action strength would be classified as a 'policy neuron' or an 'action-value neuron' by the authors' current framework, nor is it clear to me which outcome the authors would prefer. I believe the dynamics of actor-critic learning would cause the action strengths to be somewhat 'relative' (e.g. the best action is nudged toward positive strength while all others are nudged to negative strength), but I'm not sure big this effect would be, or whether this would also occur for the authors' covariance-based model, or whether this would occur if > 2 actions are available. It is possible that these types of learning tasks can't discriminate between action strengths (e.g. from an actor-critic) versus action-values (e.g. from a Q-learner). So, the authors should clarify whether they believe this is an important distinction for the present study.

2.2) Suppose we agree that neurons only count as coding the policy if they encode action probability (and not strength). Their proposed solution still seems model-dependent because it assumes that the policy is such that the probability of choosing an action is a function of the difference in action-values (*Q*_1_ – *Q*_2_) and hence policy neurons can be identified as encoding Δ*Q* and not Σ*Q*. However, there is data suggesting that humans and animals are also influenced by ratios of reward rates rather than just differences (e.g. "Ratio vs. difference comparators in choice", Gibbon and Fairhurst, J Exp Anal Behav 1994; "Ratio and Difference Comparisons of Expected Reward in Decision Making Tasks", Worthy, Maddox, and Markman, 2008). If so, then policy neuron activity could be related to a ratio (e.g. *Q*_1_ / *Q*_2_), which is correlated with both Δ*Q* and Σ*Q*.

Here is my proposed solution. It seems to me that if 'policy neurons' are equated to action probabilities, then the proper method of distinguishing policy from value coding would be to design a task that explicitly dissociates between the probability of choosing an action (encoded by policy neurons) and the action's value (encoded by action-value neurons). For instance, suppose an animal is found to choose based on the ratio of the reward rates, such that it chooses A 80% of the time when V(A) = 4*V(B). Then we can set up the following three trial types:

V(A), V(B), p(choose A)

8, 2, 80%

4, 1, 80%

4, 4, 50%

A neuron encoding V(A) should be twice as active on the first trial type as the other two trial types (since V(A) is twice as high), while a neuron encoding the policy p(choose A) should be equally active on the first two trial types (since p(choose A) = 80%). Of course, more trial types might be desired to further dissociate this from encoding of Σ*Q* and Δ*Q*. Also, note that this approach is model-dependent, because it requires a model of behavior to estimate the true p(action) on each trial (or else careful psychophysics to find two pairs of action-values that make the subject have the same action probabilities).

In general, to use this approach in a regression-based manner, one would (1) fit a model to behavior, (2) use the model to derive p(action,t) and V(action,t) for each action and each trial t, (3) fit neural activity as a function of those variables (and possibly others, such as the actually performed action, Σ*Q*, etc.), (4) test whether the neuron is significantly modulated by p(action), V(action), or both, controlling for temporal correlation using the authors' proposed method that uses task trajectories from other sessions as a control. Of course, if the model says that choice is indeed based on the value difference Δ*Q* as the authors currently assume, then this approach would simplify to the same one the authors currently propose.

Thirdly, the reviewers raised some questions about the corrections proposed and whether there in fact remained evidence for action value coding in the Basal Ganglia.

1) A critical assumption is that there exists temporal correlation strong enough to contaminate the analysis. It would be helpful to report the degree of this temporal correlation in the basal ganglia data set vs. the motor/auditory cortex data and the random walk model.

A figure, in the format of Figure 1D, showing the distribution of t-values for the actual basal ganglia data set analyzed with trial-matched Q estimates should be presented. This information is critical for effective comparisons to other data sets.

2) The authors proposed two possible solutions for this type of study. The first is to use a more stringent (and appropriate) criterion for significance, given the often wrongly assumed variance due to correlation. The permutation test is definitely in the right direction, particularly for reducing false positives. However, I am concerned by the really high rate of false negatives (~70% misses). "Considering the population of simulated action-value neurons of Figure 1, this analysis identified 29% of the action-value neurons of Figure 1 as such". Considering other unaccountable variables in typical experiments, particularly that basal ganglia neurons may have mixed selectivity both at the population and single-neuron level, such a high false negative rate seems to carry high risks of missing a true representation.

3) The authors suggested randomized blocks as the second solution. In addition to my earlier point, by their own account, such a design is not new and has been implemented in three separate studies >5 years ago. The authors pointed out some issues with those studies, which will need to be addressed in the future, but did not suggest any solutions.

4) The authors stated that the detrending analysis does not resolve the confound. However, judging from Figure 2—figure supplement 7, the detrending analysis resulted in ~29% significant Q modulation in the basal ganglia, in contrast to ~14% for random walk, ~12% for motor cortex and 10% for auditory cortex. Compared to other figures, which showed similar percentage for all four datasets, it seems that the basal ganglia data set is most robust to this analysis. Doesn't this support the idea of an action value representation in the basal ganglia?

5) The authors focus on statistical significance. Does examining the magnitude of the effects distinguish erroneous from "real" action value coding? It seems incomplete to only plot the t-values, which are important for understanding parameter precision, without presenting the parameters effect sizes. Can real action value coding be distinguished by effect sizes that were meaningfully large (i.e., substantive versus statistical significance)?

6) Along related lines, it seems like examining the pattern of effects is also useful. When comparing Figure 1D and Figure 2B, one can see that the erroneous detections included positive and negative Δ*Q* and Σ*Q* neurons, whereas for real detections (Figure 1D), there are much fewer of these neurons (by definition). All the erroneous detections generate spherical t-value plots, indicating that combinations of one or the other action value are independent. This seems not to be the case for real detections (in the authors simulations), nor in real data (Samejima et al., 2005). This suggests that any non-uniformity in detecting combinations of action value coding would be evidence that it is not erroneous (even if the type I error is not properly controlled).

7) The simulations in Figure 2 are useful, but it would be useful to translate the diffusion parameter (σ) of the random walk into an (auto) correlation. This would make it easier for a reader to interpret how this relates to real data.

8) Is the M1 data a proper control? It is hard to tell from the task description here. I wouldn't be able to replicate the task that was used given the description here. If that M1 data is published, a citation would be helpful. My concerns are whether it might have had unusually large temporal correlations and thus exaggerated the degree to which such correlations might confound action-value studies, due to either (1) having blocks of trials (as opposed to randomly interleaved trial types), (2) being a BMI task in which animals were trained to induce the recorded ensemble to emit specific long-duration activity patterns.

---

## [Author Response]

However, there were several key concerns.Firstly, the reviewers thought that there should be a more careful attention to the wider literature. In the reviewer discussion, this was thought to be of particular importance as you are making a technical point in a journal with a broad readership. It is particularly important that you are clear about which parts of the literature your results speak to.For example:1) The authors conclude that current methods of studying action-values are confounded, and propose that experiments should use actions whose reward values are not learned over time during the experiment, and instead are indicated by sensory cues with their values picked randomly on each trial.However, the authors leave unmentioned the large literature of studies taking an alternate approach: recording striatal activity during planning and execution of instructed actions for cued reward outcomes, in which each trial randomizes the instructed action, the reward, or both. The authors should discuss what implications this literature has for whether the striatum encodes action-values vs. policies vs. other variables. Examples include the work from the labs of Schultz (e.g. Hassani et al., 2001; Cromwell and Schultz, 2003; Cromwell et al., 2005), Hikosaka (e.g. Kawagoe et al., 1998; Lauwereyns et al., Neuron 2004), and Kimura (e.g. Hori et al., 2009).Most of these studies did not specifically claim that the activity they reported represents "action-values" in the reinforcement learning sense (and hence the present authors shouldn't feel obligated to try to debunk them), but they do seem highly relevant to the larger question the present authors raise. These studies did attempt to test whether neurons represented actions, values, and notably, their interaction (e.g. a cell whose activity scales more with action A's value than action B's value), which resembles the concept of "action-value".Also, these studies may be somewhat resistant to the authors' criticisms about confounds from temporal correlations (since rewards were either explicitly cued, or kept deterministic and stable in well-learned blocks of trials, rather than slowly fluctuating during extended learning) and confounds with action probability (since the actions were instructed and hence a priori equally probable on each trial).Of particular interest, a paper by Pasquereau et al. (2007) seems to fulfill all the requirements the present authors set for a test of striatal action-value coding; if so, this seems worthy of mention. That study manipulated the reward value of four actions (up, down, left, right), randomly assigning their reward probabilities on each trial and indicating them with visual cues. Unfortunately, as the present authors note, the study did not explicitly analyze their results in the action-value vs. chosen-value framework. However, the paper did report that some neurons had significant action x value interactions – for example, a cell that is more active when planning rightward movements (action), with stronger activity when the planned movement was more valuable (value), and with this value-modulation greater for rightward movements than other movements (action x value). This is not a pure chosen-value signal as the present authors seem to claim that paper reported. One could argue that it contains a key feature of an action-value signal as the value modulation is strongest for one specific action.

We agree with the reviewers that such trial designs, when trials are temporally independent, are not subject to the temporal correlation confound. We have added a paragraph about these papers and explained there why their findings cannot be used as a support to the striatal action-value representation hypothesis. In short, we do not doubt that the striatum plays an important role in decision making and learning. However, this finding, as well as the evidence in support of representation of other decision variables in the basal ganglia do not entail action-value representation in the striatum, as there are alternatives that are consistent with these findings. These points are clarified in the Discussion (Section “Other indications for action-value representation”).

Specifically regarding Pasquereau et al. (2007), we agree that the results are not consistent with pure chosen-value representation and changed the text accordingly. The finding that neurons are co-modulated by action and expected reward is indeed very interesting. However, it cannot be taken as evidence for action-value representation for several reasons. First, a policy neuron is also expected to be co-modulated by these two variables. Second, the example neurons in Figure 6 in that paper are clearly modulated by the value of *other* actions, which is inconsistent with the action-value hypothesis (no such quantitative analysis was performed at the population level). Finally, an essential test of action-value representation is that the value of the action is represented even when this action is not chosen. This was not tested in that paper (although in principle, it can be tested using existing data; The prediction of action-value representation is that the activity of that neuron is modulated by the value of the left target even when this target is not chosen). This is clarified, in short, in the “Literature search” section in the Materials and methods.

2) The authors correctly pointed out that some earlier studies of action value used a suboptimal task design and their conclusions need to be more rigorously validated. However, in the broader field, the potential risk of "drift" in neural recording has been well recognized. For example, "Neurophysiological experiments that compare activity across different blocks of trials must make efforts to be confident that any neural effects are not the result of artifacts of that design, such as slow-wave changes in neural activity over time." (Asaad et al., 2000). In the same Asaad et al. paper, a better design with repeated, alternated block types was used, similar in concept to randomized block design that the authors proposed here. Such designs have also been used in many neural studies of cognition – to name just two examples: value manipulations (Lauwereyns et al., 2002), rule manipulations (Mansouri et al., 2006). The problem thus seems relatively limited to one type of analysis that introduces temporal correlation across trials in an effort to estimate Q values. By the authors' account, this amounts to 5 papers from 3 different labs.

In response to this comment, we examined the papers proposed by the reviewer. We found that this method does not resolve the temporal correlations confound, as described in the Results section about possible solutions to the first confound (section "Possible solutions to the temporal correlations confound”) and in the Materials and methods section (the section “ANOVA tests for comparisons between blocks, excluding ‘drifting’ neurons”).

3) What about previous results arguing for prominence of a specific type of value representation? The authors touch on this, but it would be helpful to discuss specific results. In particular, the cited study of Wang et al., Nat Neuro 2003 reported that their unbiased angular measure of DMS value coding was distributed significantly non-uniformly, with net value (ΣQ) coding more prevalent than other types (their Figure 7). Whereas the null hypothesis simulations in this paper predict very different results, either a uniform distribution (Figure 2—figure supplement 8) or a dearth of ΣQ neurons (Figure 5—figure supplement 1). The authors should discuss whether this previous result can therefore still be interpreted as evidence of value coding (at least, net-value coding), rather than strictly policy coding, in the striatum.(Also, it is odd that the authors cite Wang et al. as a study that "claimed to provide direct evidence that neuronal activity in the striatum is specifically modulated by action-value", since the main result was specifically finding prevalence of net-value, not action-value coding).

We do not discuss the issue of non-uniform results in the paper but we agree that non-uniform results may be an indication of a true modulation by some variable. For example, if only neurons that are positively correlated with action-values are found (rather than negatively correlated with them) – this would be a strong indication for a modulation that is not caused by random fluctuations.

However, it is important to point out that small changes in the analysis may bias it in unexpected ways. In Author response image 1 we repeated the analysis of Wang et al., 2013 for the random-walk neurons. This analysis is slightly different form the one presented in Figure 2—figure supplement 8. There, we analyzed only the last 20 trials in each block (following Samejima et al. (2005), we now added a clarification in the figure legend). Wang et al. (2013) analyzed all the trials in a block except the first 10 and utilized 5-9 blocks. Analyzing all the trials in a block except the first 10 and utilizing 8 blocks (order of blocks as in Figure 2—figure supplement 10), surprisingly, we find a small, but significant bias towards representation of (𝛴𝑄) (p=2.9%), as in Wang et al., 2013.

Importantly, we have not fully followed the experimental setting in Wang et al. (2013). Specifically, we were not sure what was their rule for a termination of a block and we used the Samejima et al. (2005) rule. Therefore, we are unsure about the consequence of the bias we now found to their conclusions. However, this analysis shows that a biased result is not always an indication of true modulation.

With respect to the second point, we agree that Wang et al.’s (2013) main point is that the dorsomedial striatum represents net-value (i.e., 𝛴𝑄). However, they do report that "in the DMS, all categories of neuron types were represented above chance" (p. 645). Nevertheless, we added this point in the legend of Figure 2—figure supplement 8, where the Wang et al. (2013) analysis is repeated.

(Computation of p-value: The p-value for the probability of receiving this fraction of state neurons was computed under the assumption that the significant neurons were distributed uniformly between classifications. If classification is uniform, the expected fraction of neurons in each category will be 10.11%. Here we classified 11.93% of the neurons as representing state. We used 20,000 neurons in 1000 different sessions. Taking 1000 sessions as the sample size, we calculated the probability of a binomial distribution with prob. 10.11% to yield more than 119 classifications in 1000 sessions).

4) I found the authors' choice of basal ganglia data misleading (Ito and Doya, 2009). First, because these data are recordings from the nucleus accumbens and ventral pallidum, which are not the first basal ganglia structures one thinks of as encoding action values. Second, because the original authors of the study from which the data was obtained noted that action value coding was low in these structures, leading them to suggest that action value coding was not the primary function of the nucleus accumbens and ventral pallidum. This is mentioned in the Results subsection “Permutation analysis of basal ganglia neurons”, but should be noted in the Discussion (the current text in the Results could probably just be moved).

We moved the text to the Discussion (section “Temporal correlations and action-value representations”, fourth paragraph).

5) Previous methods dealing with trial correlations have different success at reducing false positive rate of detecting action-values. In particular, the method of Wang et al. (2013) comes very close to attaining the correct size of the test for action-values. Indeed, it appears to be the only existing method from which one would reasonably conclude that the ventral striatal data set analyzed probably does not exhibit much action-value coding (2-3% above the expected 5%). I think it would be useful to have a figure in the main text comparing the different methods to the authors' permutation test (using for example, just the basal ganglia data set). In addition, Wang et al.’s method is also pretty good at identifying policy neurons, which is important because it could be applied retrospectively to existing data sets.

In an attempt to make our analyses as similar as possible to the original analyses we used different thresholds for significance for different methods. Specifically, in Wang et al. analysis we find that 7% – 8% of the basal ganglia neurons represent an action value, whereas only 0.25% are expected by chance. To clarify this, we added the significance threshold to the different figures to make this difference clear.

Regarding the analysis in Wang et al. (2013) on policy neurons, we address this question in the section “Is this confound the result of an analysis that is biased against policy representation?”. This analysis indeed yields more policy than action-value neurons, but still a fraction much larger than expected by chance of policy neurons is classified as action-value neurons.

With regards to the suggestion of adding the figure, we are unsure about the added value of such a figure. In the supplementary figures we demonstrate that all these methods erroneously classify neurons in the basal ganglia recordings as representing unrelated action-values. In view of these findings, we fear that using them to identify true action-values in those recordings may mislead the readers.

6) The authors' biggest suggestion for rigorously detecting a neural action value representation is "Don't use a task with learning, use a trial-based design where subjects associate contexts with well-learned sets of action values". That is perfectly fine for scientists whose goal is specifically to test whether a brain area encodes action values. However, what about the many scientists whose explicit goal is to study neural representation of time-varying values, and hence need to use learning tasks? Many scientists are studying (1) the neural basis of value learning, (2) brain areas specifically involved in early learning (not well-trained performance), (3) motivational variables specifically present for time-varying action values not well-trained ones (e.g. volatility, certain forms of uncertainty, etc.). If the authors can give an approach that will let these scientists make accurate estimates of neural time-varying value coding during learning tasks, that would certainly be valuable to the field.I feel like their methods could potentially be used to achieve this in a straightforward way (by combining their novel permutation test from Part 1 of their paper with their method of testing for correlation with both sum and difference of values from Part 2 of their paper). But they don't lay this out explicitly in their paper at present, since they are more focused on the narrower implication ("Do striatal neurons encode action values?") rather than the broader implication of their results ("In general, how can one properly measure time-varying action values?").

The paper addresses two confounds, that are somewhat orthogonal. The temporal correlation confound can be addressed using the permutation analysis of Figure 3, which can provide strong support to the claim that the activity of a particular neuron co-varies with learning. This is a general solution for scientists studying slow processes such as learning.

Precisely defining or interpreting what the activity of the neuron represents (for example an action-value or policy) is more challenging and in general, there are no easy solutions and caution should be applied when interpreting the data. We now discuss these points in the 'Temporal correlations – beyond action-value representation' section of the Discussion.

With respect to the proposed solution, to rule out policy representation, the analysis in Figure 6 includes a regression on an orthogonal variable – state. For the two variables to be orthogonal it is required mathematically that the two action-values will have the same variance (section “A possible solution to the policy confound”). This can be achieved in a controlled experiment where reward probabilities are used, but we cannot control for the variance of the action-values when we estimate them from behavior. Therefore, we could not find a way to combine the solution from Figure 3 with the regression analysis from Figure 6. However, in other cases, this may not be an issue, depending on the specific variable and question.

Secondly, the reviewers had some particular concerns about the action value vs. policy representation issue. For example:1) Regarding the second confound of policy vs. action value:- The authors seem to be arguing against a straw-man version of how to relate neural activity to behavior. Typically we infer the underlying computations by testing how well different hypothesized models can fit the behavior and then search for correlates of the most likely computation in the brain. The authors seem to test only how well the neural activity correlates with different hypothesized models.

We respectfully disagree with the review for two reasons:

First, the reviewer hints that because action-value based models best describe behavior, we should search for action-value representations. We would like to note that while the view that action-value based models best describe behavior is widespread, there is strong evidence that favors other models (e.g., Erev et al., Economic Theory, 2007, see also Shteingart and Loewenstein, 2014 for review). Therefore, it is still an open question whether action-value representation exists in the brain.

Second, policy representation (representation of the probability of choice) is likely to exist even if the brain computes action-values. If neurons represent policy, then they may be misclassified as representing action-values.

- The proposed solution for distinguishing policy from action value also has a very high rate of false negative (78%).

We agree with this point and we remedied the analysis to decrease its false negative rate. For true action-value neurons, the rate of correct detection vs. false negatives depends on the strength of their modulation by action-value, together with the power of the analysis.

We used neurons whose correct detection rate in the original analyses was comparable to the literature (~40%). The analysis in the previous version of the manuscript decreased this rate to 22%. It indeed suffered from limited power also because it only employed 80 trials. To increase the power of the analysis, we repeated the analysis using 400 trials in total (rather than the original 280 trials) and conducting the analysis on the last 200 trials. We now correctly classify 32% of action-value neurons as such (see Figure 6). Considering that the original analysis in Figure 1 was biased towards classifying neurons as representing action-value, rather than policy or state and that our new analysis requires passing two significance tests, we take this correct detection rate to be reasonable.

We changed Figures 4 and 6, together with their figure legends and descriptions of the analysis accordingly.

2) I feel that the point the authors make about action-value vs. policy representations may actually be underselling the true extent of the confound, and so their proposed solution may not be sufficient. However, this all depends on how the authors want to define a 'policy neuron' vs. a 'value neuron', as I explain below. I think they should clarify this.2.1) Their arguments seem to assume that neural policy representations are in the form of action probabilities, which can then be identified by the key signature that they relate to action-values in a 'relative' manner (e.g. an 'action 1' neuron that is correlated positively with Q_1_ must be correlated negatively with Q_2_), and hence will be best fit as encoding ΔQ (Q_1_ – Q_2_). However, depending on how they define 'policy', this may not be the case.Notably, even for reinforcement learning agents that do not explicitly represent action-values, few of them directly learn a policy in its most raw form of a set of action probabilities. Instead, they represent the policy in a more abstract parameter space. The simplest parameterization is a vector of action strengths, one for each possible action. Then during a choice, the probability of each action is determined by applying a choice function (e.g. softmax) to the action strengths of the set of actions that are currently available. The choice's outcome is then used to do learning on the action strengths. This method is used by some traditional actor-critic agents (which represent state values and action strengths, but not action-values). My impression is that the authors' covariance-based model is similar, in that the variables that it updates when it learns are the input weights W_1_ and W_2_ to each pool (with one input weight for each action, thus being analogous to action strengths).Note that in these models, the action strengths are not explicitly represented in a 'relative' manner; only the resulting action probabilities are (since the probabilities must sum to 1). It's not clear to me whether a neuron encoding an action strength would be classified as a 'policy neuron' or an 'action-value neuron' by the authors' current framework, nor is it clear to me which outcome the authors would prefer. I believe the dynamics of actor-critic learning would cause the action strengths to be somewhat 'relative' (e.g. the best action is nudged toward positive strength while all others are nudged to negative strength), but I'm not sure big this effect would be, or whether this would also occur for the authors' covariance-based model, or whether this would occur if > 2 actions are available. It is possible that these types of learning tasks can't discriminate between action strengths (e.g. from an actor-critic) versus action-values (e.g. from a Q-learner). So, the authors should clarify whether they believe this is an important distinction for the present study.

The reviewer is making an interesting and important point. An initial requirement for a neuron to be considered an action-value neuron, a policy neuron or any decision variable-neuron, is that it is significantly more correlated with these decision variables than with decision variables that are unrelated to the current task. The permutation analysis of Figure 3 can be used to find such neurons.

The question of which decision variable the neuron represents (assuming that it passed the permutation test) is a more difficult one. The reason is that the different decision variables are correlated. Moreover, because these variables are all some function of past actions and rewards, and relate to future choice, many existing and future decision-making models are expected to have modules whose activity correlates with these variables. One may argue that the question of whether neurons represent action-value, policy, state or some other correlated variable is not an interesting question. This is because all these correlated decision variables implicitly encode action-value. Even direct-policy models can be taken to implicitly encode action-value, because policy is correlated with the difference between the action-values. However, we believe that the difference between action-value representation and representation of other variables is an important one, because it centers on the question of the computational model that underlies decision-making in these tasks.

Often, reports of action-value representation are taken to support the hypothesis that action-values are explicitly computed in the brain, and that these action-values play a specific role in the decision making process. While other models may include no such calculation they can still include neuronal activity that correlates with action-value, as in the covariance-based plasticity model (at the level of the population). One proper way of ruling out competing hypotheses about the variables the neuronal activity correlates with is to test for significant correlations in directions that are correlated with action-value but are orthogonal to each of the competing hypotheses.

Clearly, one cannot attempt to rule out all possible hypotheses. However, even in the restricted framework of value-based Q-learning, a necessary condition for a neuron to be considered as representing an action-value is that it is not representing other decision variables *of that model* such as policy. Regarding alternative models for learning, clearly the more restrictive the characterization of the response properties of a neuron in the task, the more informative it is about the underlying neural computation.

We added a section in the Discussion titled “Are action-value representations a necessary part of decision making? “that addresses these issues.

2.2) Suppose we agree that neurons only count as coding the policy if they encode action probability (and not strength). Their proposed solution still seems model-dependent because it assumes that the policy is such that the probability of choosing an action is a function of the difference in action-values (Q_1_ – Q_2_) and hence policy neurons can be identified as encoding ΔQ and not ΣQ. However, there is data suggesting that humans and animals are also influenced by ratios of reward rates rather than just differences (e.g. "Ratio vs. difference comparators in choice", Gibbon and Fairhurst, J Exp Anal Behav 1994; "Ratio and Difference Comparisons of Expected Reward in Decision Making Tasks", Worthy, Maddox, and Markman, 2008). If so, then policy neuron activity could be related to a ratio (e.g. Q_1_ / Q_2_), which is correlated with both ΔQ and ΣQ.

We agree, but any analysis can only consider and compare the hypotheses that are explicitly acknowledged. We added a paragraph in the Discussion addressing this point (section “Differentiating action-value from other decision variables”, fifth paragraph).

Here is my proposed solution. It seems to me that if 'policy neurons' are equated to action probabilities, then the proper method of distinguishing policy from value coding would be to design a task that explicitly dissociates between the probability of choosing an action (encoded by policy neurons) and the action's value (encoded by action-value neurons). For instance, suppose an animal is found to choose based on the ratio of the reward rates, such that it chooses A 80% of the time when V(A) = 4*V(B). Then we can set up the following three trial types:V(A), V(B), p(choose A)8, 2, 80%4, 1, 80%4, 4, 50%A neuron encoding V(A) should be twice as active on the first trial type as the other two trial types (since V(A) is twice as high), while a neuron encoding the policy p(choose A) should be equally active on the first two trial types (since p(choose A) = 80%). Of course, more trial types might be desired to further dissociate this from encoding of ΣQ and ΔQ. Also, note that this approach is model-dependent, because it requires a model of behavior to estimate the true p(action) on each trial (or else careful psychophysics to find two pairs of action-values that make the subject have the same action probabilities).In general, to use this approach in a regression-based manner, one would (1) fit a model to behavior, (2) use the model to derive p(action,t) and V(action,t) for each action and each trial t, (3) fit neural activity as a function of those variables (and possibly others, such as the actually performed action, ΣQ, etc.), (4) test whether the neuron is significantly modulated by p(action), V(action), or both, controlling for temporal correlation using the authors' proposed method that uses task trajectories from other sessions as a control. Of course, if the model says that choice is indeed based on the value difference ΔQ as the authors currently assume, then this approach would simplify to the same one the authors currently propose.

This is an elegant experimental design and not unlike the one we consider in Figure 6. However, with respect to the proposed analysis, there are two important differences. One is the question of whether behavior is modulated by the ratio of reward rates, the difference of reward rates or a different function. In the paper we posited that it is the difference in the reward rates that modulates behavior when analyzing the data in the value-based framework. We agree, that it is possible that the ratio is a better predictor of behavior. Our choice followed that of the previous publications and is based on the assumption of the Q-learning model that the probability of choice is a monotonic function of the difference between action-values.

Second, in point 4, the reviewers propose to test the type of representation by looking for significant modulation or the lack of it. However, a non-significant result for one variable, is not an indication that it was not the modulator. As described in Figure 5, this can lead to confounds. Furthermore, policy and action-value will have shared variance, and so some of the modulation of the neuronal activity cannot be conclusively attributed to any of them. Therefore, it is better to use model comparison (likelihood) when considering the results of this analysis. In our manuscript we focus on significance tests that can rule out specific possibilities under the null hypothesis.

Note, that the design suggested by the reviewers can also be used to reject the hypothesis that neurons are policy neurons. For neurons whose activity differs significantly between the first two cases (p(choose A)=80%) the null hypothesis that they represent policy can be rejected. In the experimental design we simulate in the paper (Figure 6) this is like comparing the activity of neurons at the end of two blocks where the policy is similar (this is an assumption which can be tested empirically). We can compare the neural activity in (0.1, 0.5) with (0.5, 0.9), and the activity in (0.5, 0.1) with (0.9, 0.5). To rule out the possibility of state representation we should compare the activity and the end of the following blocks: compare (0.1, 0.5) with (0.5, 0.1), and (0.5, 0.9) with (0.9, 0.5). As the reviewers note, this is in fact exactly what we do in the analysis in Figure 6. We regress neuronal activity on state – sum(0.1, 0.5)=0.6, sum(0.5, 0.9)=1.4, sum(0.5, 0.1)=0.6, sum(0.9, 0.5)=1.4. This effectively compares activity in cases with the same policies in a regression model.

Thirdly, the reviewers raised some questions about the corrections proposed and whether there in fact remained evidence for action value coding in the Basal Ganglia.1) A critical assumption is that there exists temporal correlation strong enough to contaminate the analysis. It would be helpful to report the degree of this temporal correlation in the basal ganglia data set vs. the motor/auditory cortex data and the random walk model.

Author response image 2 shows a plot of the autocorrelation of the spike counts in each trial for the different data sets (averaged over the spike counts in each group; light-colors denote SEM; computed using MATLAB’s ‘autocorr’ function).

We believe that it is better to refrain from including this figure in the paper for two reasons: (1) The autocorrelations relevant for the temporal correlations confound are those associated with the time-scale relevant for learning, tens of trials. Computing such autocorrelations in experiments of a few hundreds of trials introduces substantial biases (Newbold and Agiakloglou, 1993; Kohn, 2006). This is also demonstrated in the negative autocorrelation of the random-walk spike counts, computed using sessions of 151-379 trials. Alternative measures for autocorrelation are also problematic when applied to small samples, see (Kohn, 2006). (2) We are not aware of theoretical mapping from the autocorrelation function to the temporal correlations confound. For example, considering the autocorrelations below, it is not clear how to compare the basal ganglia and the motor cortex datasets with respect to the temporal correlations confound when considering their autocorrelation functions. For these reasons, computing autocorrelation functions to quantify the temporal correlations confound may be misleading rather than useful.

We added a paragraph to the manuscript, describing the potential problems with the autocorrelation measure (section “Possible solutions to the temporal correlations confound”, second paragraph).

**Author response image 2. respfig2:** 

A figure, in the format of Figure 1D, showing the distribution of t-values for the actual basal ganglia data set analyzed with trial-matched Q estimates should be presented. This information is critical for effective comparisons to other data sets.

We added Figure 3—figure supplement 1, which reports this information.

2) The authors proposed two possible solutions for this type of study. The first is to use a more stringent (and appropriate) criterion for significance, given the often wrongly assumed variance due to correlation. The permutation test is definitely in the right direction, particularly for reducing false positives. However, I am concerned by the really high rate of false negatives (~70% misses). "Considering the population of simulated action-value neurons of Figure 1, this analysis identified 29% of the action-value neurons of Figure 1 as such". Considering other unaccountable variables in typical experiments, particularly that basal ganglia neurons may have mixed selectivity both at the population and single-neuron level, such a high false negative rate seems to carry high risks of missing a true representation.

The rate of misses of action-value neurons in our analysis depends on the parameters that we used to model these neurons. We used parameters such that the "standard" methods miss approximately 60% of the action value neurons. With the permutation test we miss approximately 70%. Other parameters would yield different rates of misses. If selectivity is weak then indeed, it will be more difficult to identify such neurons. However, a necessary condition for a neuron to be classified as a task-related neuron is that it is more correlated with decision variables in its corresponding session than with these decision variables in other sessions. We do not see a way around it even if this requirement is associated with a substantial rate of false identifications.

One approach to increase the power of any analysis will be to use as many trials as possible, as can be seen from the increase in the correct detection rate in Figure 6, caused by the addition of trials (we could not add trials in this analysis because we analyzed the original neurons of Figure 1). Another alternative is to consider population coding rather than to focus on individual neurons. This analysis is, however, beyond the scope of this paper.

3) The authors suggested randomized blocks as the second solution. In addition to my earlier point, by their own account, such a design is not new and has been implemented in three separate studies >5 years ago. The authors pointed out some issues with those studies, which will need to be addressed in the future, but did not suggest any solutions.

We are not sure that we understand this comment. In our second solution we proposed randomized trials and not randomized blocks. If the reviewer relates to the similarity of our second solution to (Fitzgerald, Friston and Dolan, 2012) then crucially, we used reward probabilities in the analysis and not estimated action-values. This removes temporal correlations which are present when estimated action-values are used (see Figure 2—figure supplement 9). In addition, our analysis in Figure 6 rules out policy and state representations, which was not present in (Fitzgerald, Friston and Dolan, 2012). This last point is also relevant to (Cai et al., 2011 and Kim et al., 2012).

4) The authors stated that the detrending analysis does not resolve the confound. However, judging from Figure 2—figure supplement 7, the detrending analysis resulted in ~29% significant Q modulation in the basal ganglia, in contrast to ~14% for random walk, ~12% for motor cortex and 10% for auditory cortex. Compared to other figures, which showed similar percentage for all four datasets, it seems that the basal ganglia data set is most robust to this analysis. Doesn't this support the idea of an action value representation in the basal ganglia?

Originally, we were not clear enough on this issue. We’ve added clarifying sentences in the figure legends. The analysis of the basal ganglia data in Figure 2—figure supplement 7*erroneously* identified *unrelated* action-values from simulations. In fact, this analysis indicates that detrending is even *less* useful there than in other datasets.

5) The authors focus on statistical significance. Does examining the magnitude of the effects distinguish erroneous from "real" action value coding? It seems incomplete to only plot the t-values, which are important for understanding parameter precision, without presenting the parameters effect sizes. Can real action value coding be distinguished by effect sizes that were meaningfully large (i.e., substantive versus statistical significance)?

To address this comment, we compared the explained variance of the action-value and random-walk neurons used in our paper. Surprisingly, the explained variance of the random-walk neurons is *higher* than that of the true action-value neurons.

**Author response image 3. respfig3:** 

One may argue that very high explained variance (say R^2^ > 0.25) can be used as conclusive evidence of action-value representation. However, we find that if the diffusion coefficient of the random-walk neurons is sufficiently large then a substantial fraction of the neurons will exhibit high values of R^2^. For example, with a diffusion coefficient of 0.5 31% of the random-walk neurons exhibit R^2^ > 0.25.

6) Along related lines, it seems like examining the pattern of effects is also useful. When comparing Figure 1D and Figure 2B, one can see that the erroneous detections included positive and negative ΔQ and ΣQ neurons, whereas for real detections (Figure 1D), there are much fewer of these neurons (by definition). All the erroneous detections generate spherical t-value plots, indicating that combinations of one or the other action value are independent. This seems not to be the case for real detections (in the authors simulations), nor in real data (Samejima et al., 2005). This suggests that any non-uniformity in detecting combinations of action value coding would be evidence that it is not erroneous (even if the type I error is not properly controlled).

We partially answer this question (question 3 in the first set of comments above). Some non-uniformities may indeed indicate that the result are not due to random modulations. However, even when dealing with random modulations we may see certain biases that are caused by the design of the analysis. Another example is Figure 2. There we find that in the random-walk dataset, the fraction of state neurons is larger than that of policy neurons. We shortly address the fact that the results may be biased towards a specific classification in some experimental designs in Figure 2—figure supplements 4, 5, and Figure 3—figure supplement 1.

7) The simulations in Figure 2 are useful, but it would be useful to translate the diffusion parameter (σ) of the random walk into an (auto) correlation. This would make it easier for a reader to interpret how this relates to real data.

As discussed above, we fear that presenting autocorrelations may be misleading. Particularly, the autocorrelations of the random-walk function for a finite (and small) number of trials, which is relevant for experiments is very different from the function obtained when the number of trials is large. This is depicted in Author response image 4, where we compare the autocorrelation of the random-walk sessions of the paper, with the autocorrelation function of the same process, computed using 5,000 trials.

**Author response image 4. respfig4:** 

8) Is the M1 data a proper control? It is hard to tell from the task description here. I wouldn't be able to replicate the task that was used given the description here. If that M1 data is published, a citation would be helpful. My concerns are whether it might have had unusually large temporal correlations and thus exaggerated the degree to which such correlations might confound action-value studies, due to either (1) having blocks of trials (as opposed to randomly interleaved trial types), (2) being a BMI task in which animals were trained to induce the recorded ensemble to emit specific long-duration activity patterns.

The motor cortex data was recorded in Eilon Vaadia’s lab and has not been published yet. We agree that the specific task the subject is performing may influence the overall firing rate or the temporal correlations in the neural activity and hence the false positive rates in the detection of action-value representation. However, we think it is unlikely that the recordings in this data set are an outlier in terms of autocorrelations. First, the monkey was extensively trained and all trials were identical, so there is nothing in the design of the task that suggests long-term correlations between trials. Second, the monkey was conditioned to enhance the power of beta band frequencies (20-30Hz). This frequency band is two orders of magnitude different than the time scale separating different trials (on average 14.2 seconds). Finally, we considered spike count *prior* to the beginning of the trials, while the monkey was still waiting for a GO signal.